# IAGENT: A WEB SEARCH FRAMEWORK FOR NOISE ISOLATION AND EXTENDED INFORMATION ACCESS

## ABSTRACT

Large language models (LLMs) have demonstrated remarkable capabilities in tool use and information integration, yet their fixed context windows pose significant challenges for conducting deep, web-scale research. Although many web search systems exist today, they still face substantial limitations when applied to research scenarios that require aggregating and synthesizing large volumes of information. To address these issues, we introduce IAgent, a new deep-research framework. Our approach employs three specialized agents that collaboratively gather extensive, up-to-date information from multiple sources. We further redesign several key components of the pipeline and decouple them as much as possible to ensure the validity, reliability, and accuracy of the final information delivered to the LLM. Experimental results show that our framework achieves strong performance across multiple deep-research benchmarks, demonstrating its effectiveness and robustness in handling complex information-intensive tasks.

## 1 INTRODUCTION

Large Language Models (LLMs) have rapidly evolved from simple predictors into powerful autonomous agents capable of multi-step reasoning, tool use, and complex planning (Java et al., 2025; Guo et al., 2024). These agent frameworks often employ iterative planning and execution loops (Yao et al., 2023; Significant Gravitas, 2023), and PlanBench frameworks (Valmeekam et al., 2023), augmented by memory mechanisms for long-horizon tasks (Packer et al., 2024; Xu et al., 2025). For existing web search agent frameworks, manager–worker communication patterns (Zhang et al., 2024) are commonly adopted. The manager devises high-level strategies and decomposes tasks, while the worker gathers relevant information for each subtask. Both the manager and the workers operate under the *thought–action–observation* paradigm (abbreviated as **T**, **A**, and **O**, respectively). Specifically, the manager formulates a plan (T), then invokes its own tools or the workers (A), and after receiving feedback (O), it continues iterating until it deems that a final answer has been obtained. For the worker, it first generates a query (T), then calls a tool to search for relevant webpages (A), obtains search results (O), selects the most relevant page (T), browses it (A), and acquires information (O), repeating this loop until enough evidence is collected to report back to the manager.

While these frameworks have achieved significant success, substantial limitations persist when they operate within open network environments, especially for deep research (see Section 2.1) tasks that require agent frameworks to maintain extensive contextual states. To understand the limitations of current frameworks, we identify three critical design flaws in existing agent frameworks: (i) **High Noise**: The noise arises in two stages. As shown in Figure 1(a), the first is the **search noise** generated by the worker. Open network environments contain an overwhelming amount of heterogeneous information, and the sensitivity of web search tools (e.g., Google Search) means that even subtle variations in search queries may result in the omission of relevant or correct information. Intuitively, the query in Case 1 seems more accurate, but in reality, only the query in Case 2 yields search results containing webpages with the correct answer. One must first locate the webpages corresponding to articles published in 2020 and then perform further browsing to extract the answer. The former query contains too many keywords, which weakens the importance of the term "2020" and "article". After an LLM goes down an incorrect path, it could have rejected this path in time by reading webpage abstract. However, the LLM tends to reinforce its prior choices (Zhang et al., 2025; Gan et al., 2025), thereby often browsing webpages that appear relevant but are actually irrelevant. We refer to this tendency as **inertia bias** (see Appendix A). If reasoning continues along irrelevant webpages,

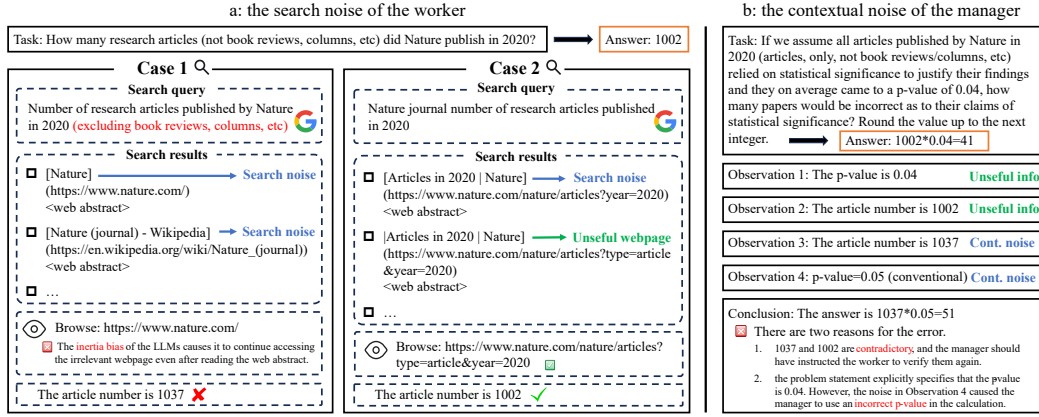

Figure 1: Two kinds of noise. The words in green indicate useful information, and the words in blue indicate noise. The words in red reveal the causes of the two types of noise. In subfigure (a), the <web abstract> denotes the summary information of webpages returned by Google search. For two seemingly similar queries, the results differ drastically. In Case 1, due to the inherent inertia bias of the LLMs, the worker continues searching along the query, and arrives at an incorrect answer. In contrast, in Case 2 the query is shorter, which makes the keywords "2020" and "article" more salient, thereby enabling the correct webpage to be located. In subfigure (b), we omit the manager's thought and action steps, retaining only the observation results. The contextual noise misleads the manager into faulty reasoning, ultimately yielding an incorrect answer.

it may at best increase contextual overhead and waste action steps, or at worst lead to entirely incorrect answers. Figure 1(b) illustrates the second type, namely **contextual noise** generated by the manager. Since the information returned by each of its actions is not always directly useful for the final answer, as the number of iterations grows, increasingly long contexts and accumulated noise may cause the manager to hastily output an incorrect answer. More frustratingly, the effect of inertia bias is sufficiently strong that the mere addition of filtering modules at these two stages cannot resolve the problem. (ii) **Monolithic Information Acquisition**: Most frameworks primarily rely on web browsing as the sole means of information gathering. In most cases, this method is effective. However, for deep research tasks, when batch queries are required, traditional web browsing is insufficient to meet efficiency requirements. *For instance, consider a task that requires analyzing the revision history of a large set of Wikipedia articles. To collect such data, one would need to repeatedly access the edit histories of hundreds or even thousands of pages, which is inefficient.*

To address these issues, we propose Isolation Agent (IAgent), a web search agent framework that focuses on noise isolation and extended information access. Consistent with established manager–worker communication patterns, our design incorporates a ManagerAgent serving as the manager and a SearchAgent serving as the worker. Beyond this conventional setup, we introduce three major improvements, specifically: (i) For SearchAgent, we propose the context-isolated filtering module. This module exclusively takes as input a triplet consisting of a question, the latest plan, and candidate webpages. It then leverages LLMs' contextual reasoning capabilities to perform a fine-grained relevance evaluation. This context-isolated filtering mechanism can effectively filter out irrelevant webpages at an early stage, thereby preventing the SearchAgent from generating **search noise** due to the LLM's inertia bias. (ii) For ManagerAgent, we propose the isolation-based stepwise validation module. When the ManagerAgent deems an answer ready, this module first extracts the reasoning chain leading to that answer from the ManagerAgent's memory, and then conducts a step-by-step review and validation. Finally, the module aggregates the feedback from the context-isolated review and validation stage, providing a reliable decision-making basis for subsequent actions. Therefore, it addresses the **high contextual noise** problem of the ManagerAgent. (iii) We further introduce CoderAgent, which also acts as the worker. It is dedicated to solving complex mathematical problems and programmatically interacting with external services. It can leverage loop structures in code to automate large-scale data processing, and it can further access structured data via unauthenticated

APIs. *For instance, API-based access enables the efficient retrieval of Wikipedia revision histories in bulk.* In this way, the problem of **monolithic information acquisition** is successfully solved.

We take smolagents DR (Roucher et al., 2025b) as our initial baseline and build improvements upon it. Our experimental results demonstrate that IAgent achieves a significant performance enhancement over the smolagents DR baseline on the GAIA benchmark (Mialon et al., 2023) under similar model configurations. Notably, by blocking access to irrelevant webpages in advance, our method reduces token cost by 33% compared with smolagents DR. With stronger model configurations, IAgent achieves SOTA performance on both GAIA and WebWalker benchmarks among open-source frameworks.

## 2 RELATED WORK

### 2.1 DEEP RESEARCH AGENTS

Deep research refers to conducting multi-step searches on the internet for complex tasks (OpenAI, 2025). These skills are similar to the requirements for Artificial General Intelligence (AGI), and have therefore attracted a large and growing number of researchers. For existing deep research agent frameworks, manager–worker communication patterns (Zhang et al., 2024) are commonly adopted. For the manager, reducing hallucinations and noise is crucial for improving system capability. For example, KnowAgent (Zhu et al., 2025b) introduces an action-knowledge base to constrain action path generation, Agent KB (Tang et al., 2025) utilizes a teacher model to guide the student model in searching and providing suggestions, and Agent Workflow Memory (Wang et al., 2024b) reuses summarized sub-workflows. However, all of these methods rely on external experience databases. For the worker, ASCoT (Zhang et al., 2025) and Slow-Thinking (Gan et al., 2025) intervene at critical steps, while WebShaper (Tao et al., 2025) employs compositional Knowledge Projection (KP) operations to finely control the reasoning structure. Their common objective is to detect issues as early as possible and correct the action trajectory.

### 2.2 ACTION MODULE IN LLM AGENTS

With the improvement of LLMs' context length and comprehension, researchers have begun to build them into agent systems (Significant Gravitas, 2023), where agents iteratively plan and execute to solve complex tasks. ReAct (Yao et al., 2023) couples reasoning with tool use, while Plan-Bench (Valmeekam et al., 2023) and FlowBench (Xiao et al., 2024) provide structured environments for stepwise planning. When taking actions, traditional agents often generate structured text (Park et al., 2023; Schick et al., 2023) or JSON (Qin et al., 2023; Chase, 2022) to call tools. Multi-agent systems even rely on such representations for communication among individual agents (langchain ai, 2023; Wu et al., 2024). However, these forms restrict action expressiveness and lack flexibility. Code Act (Wang et al., 2024a) represents actions using executable code to better elicit LLM capabilities, and smolagents (Roucher et al., 2025a) employs Code Act to enable more efficient communication and tool invocation in multi-agent systems. However, smolagents is not specifically designed for code-driven deep web browsing, which leaves room for further optimization in deep research tasks.

## 3 METHODOLOGY

### 3.1 WORKFLOW FRAMEWORK

We design the IAgent framework based on the smolagents library (Roucher et al., 2025a). In conventional methods, LLM agents typically generate actions through text (Park et al., 2023; Schick et al., 2023) or JSON (Qin et al., 2023; Chase, 2022). However, these methods suffer from two major limitations. The action space is restricted by a predefined set of tools, limiting expressiveness, and the lack of flexible control flow makes it difficult to process tasks in batches. In contrast, by leveraging smolagents, abstracting actions as Python code not only overcomes these limitations but also allows the system to reuse existing mature libraries (such as bs4, xml, etc.). Consistent with established manager–worker communication patterns (Zhang et al., 2024), our design incorporates a ManagerAgent serving as the manager and a SearchAgent serving as the worker. Beyond this

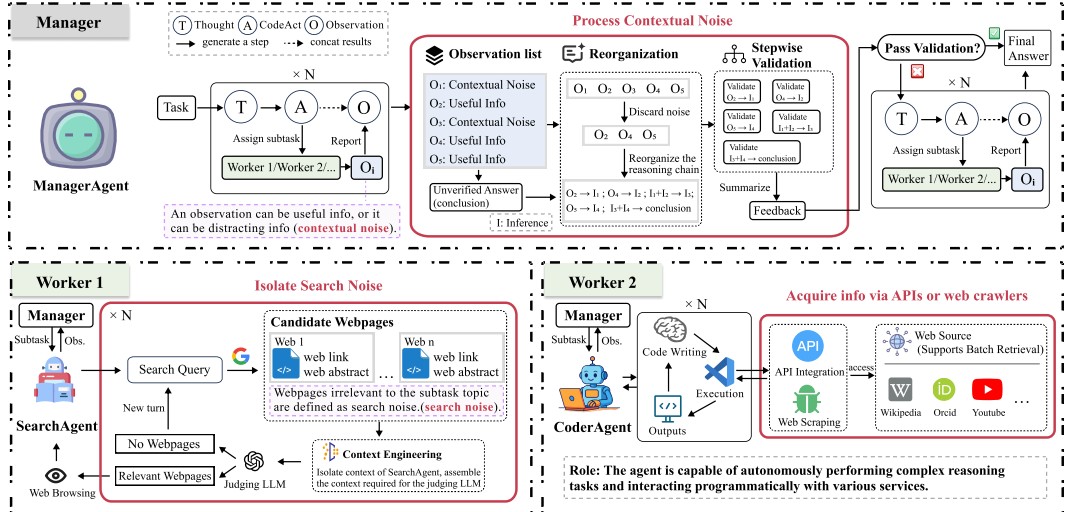

Figure 2: Framework of IAgent. The three components highlighted in red in the figure constitute the main focus of our optimization. **Manager**: ManagerAgent alternates between thought, codeact, and observation. The actions here include invoking Work 1 (SearchAgent), Work 2 (CoderAgent), or calling equipped tools. It continuously accumulates contextual noise during iteration. When ManagerAgent deems the answer is obtained, it will automatically trigger its validation module (see Section 3.3). When it passes validation, it outputs the answer; otherwise, it continues to iterate. **Work 1**: SearchAgent is responsible for collecting information from the web. We specifically optimized the Filter stage (see Section 3.2) to prevent search noise generated during searching. Once collecting enough information, SearchAgent will organize and report it. **Work 2**: CoderAgent leverages APIs or crawler tools to batch retrieve information or access content protected by JavaScript (See Appendix B.1 for API registration and usage details.). It also undertakes certain reasoning tasks when necessary. Once collecting enough information, CoderAgent will organize and report it.

conventional setup, we design a validation module for the ManagerAgent (see Section 3.3) and a filter module for the SearchAgent (see Section 3.2). Finally, we introduce the CoderAgent as part of IAgent to strengthen its overall capability. Figure 2 illustrates the overall framework of IAgent.

**ManagerAgent.** This agent undertakes responsibility for overarching coordination, encompassing global planning, task allocation, information integration, and the validation of result correctness. It is driven by an LLM, which exploits its intrinsic logical reasoning capabilities to analyze and decompose complex tasks, rather than adhering to rigid workflows. ManagerAgent is an extension of ReAct. Before each action, it first performs reasoning, then writes code to represent the action, and finally the system automatically parses it and delegates the task to other tools or team members.

**SearchAgent.** This agent undertakes the majority of tasks related to web search. We equip it with multiple tools to enable more effective information retrieval from the internet. It is capable of processing most online resources and can also leverage archival services to retrieve historical content. SearchAgent also follows the ReAct paradigm: it first performs reasoning, then selects an appropriate tool from its toolkit and generates the corresponding arguments in real time. After waiting for the tool's output, it proceeds to the next action, iterating in this manner until sufficient information has been collected. For detailed tool information, please refer to Appendix B.2.

**CoderAgent.** While the aforementioned dual-agent configuration is sufficient for many web search scenarios and aligns with most system architectures (Zhu et al., 2025a; Roucher et al., 2025a; Pang et al., 2025), we have identified significant limitations when applied to deep research. The SearchAgent, which relies on direct access to and parsing of webpages, operates inefficiently when required to integrate data across multiple sites. More critically, it often fails when confronted with content dynamically rendered by JavaScript or information protected by APIs, and it lacks sufficient capabilities for structured data extraction. To address these shortcomings, we introduce the CoderAgent, which focuses on advanced programming and complex reasoning tasks. When the SearchAgent is

unable to retrieve necessary information, the CoderAgent can programmatically interact with various services by generating code to call APIs or control browser automation tools, thereby extending the system's functional boundaries. This integration is seamless, as our framework natively employs code as its action representation. Moreover, beyond information acquisition, the CoderAgent can independently undertake sophisticated reasoning tasks, transferring potentially multi-round deliberation into a separate context. This design substantially raises the upper bound of IAgent in handling challenging problems, while simultaneously reducing unnecessary context overhead.

## 3.2 CONTEXT-ISOLATED FILTER MODULE

Search is the beginning of all web browsing. Analogous to human behavior, the SearchAgent needs to perform web searches to retrieve relevant webpage links.

In the workflow of the tool `web_search`, the SearchAgent first generates query statements. Subsequently, relevant webpage links and abstracts are retrieved through Google Search. Owing to the stochastic nature of natural language generation, a single question can be mapped to semantically divergent queries, occasionally yielding results that are irrelevant to the intended task. In the absence of any filter module, the SearchAgent will invariably select at least one invalid URL for access. Moreover, due to the inherent inertia bias (i.e., the tendency of an LLM to persist along its own proposed query path, see Appendix A for a detailed explanation), naively integrating a filter module into the existing workflow fails to resolve the issue. The effect of this issue is substantial. Directly utilizing such non-pertinent information for subsequent web navigation wastes computational resources and risks misleading the reasoning process with extraneous information.

To address this, we designed a context-isolated filter module that performs intelligent screening of candidate search results before webpage access. The key point of this module is its deliberate isolation of the evaluation process from the main workflow of the SearchAgent. It exclusively takes as input a triplet consisting of a question, the latest plan, and candidate webpages, leveraging LLMs' contextual reasoning capabilities to perform a fine-grained relevance evaluation. This context-free design ensures that each evaluation is independent and objective, completely eliminating potential interference from the SearchAgent's historical behaviors, such as the most recent query. When relevance is insufficient, the module triggers an adaptive query rewrite, thereby significantly improving retrieval efficiency and accuracy. Figure 3 illustrates the workflow of this module.

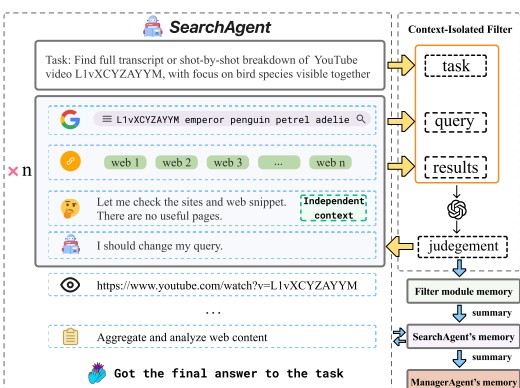

Figure 3: Workflow of the context-isolated filter module. It extracts the task, query, and result from the SearchAgent's memory and evaluates their relevance after each query.

## 3.3 ISOLATION-BASED STEPWISE VALIDATION MODULE

When the agent considers an answer to be obtained, existing methods either directly output the answer (Roucher et al., 2025a) or submit the entire reasoning process to an LLM for validation (Wu et al., 2025b). However, these methods are often ineffective. The historical context contains substantial irrelevant information that can interfere with validation. Furthermore, due to the sycophantic tendencies of LLMs, the proposed answer is often confirmed regardless of its correctness.

To address these issues, we propose an optimized validation workflow. When the ManagerAgent deems an answer ready, it automatically enters the answer validation module, which operates in two stages. We implement this component as a ValidationAgent, while conceptually it remains a module within the overall framework. The operational mechanism of this module is illustrated in Figure 4.

**Stage 1: Reasoning Process Reorganization.** In the first stage, the ValidationAgent extracts key reasoning steps from the ManagerAgent's execution memory, including the original task description and all tool-call sequences, and then employs a specialized prompt template to organize these

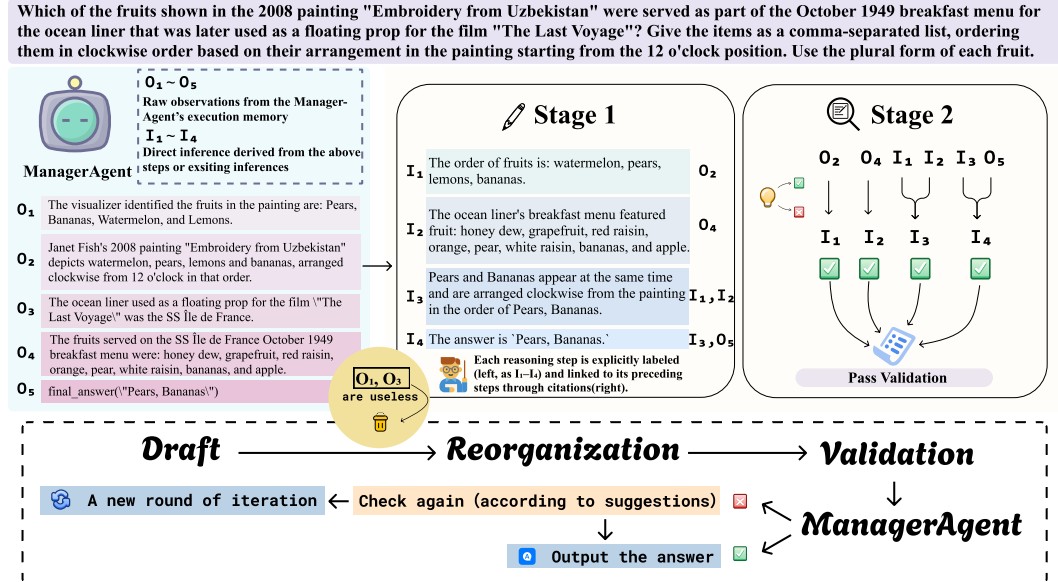

Figure 4: Isolation-based stepwise validation module. This module is automatically triggered in the form of a ValidationAgent when the ManagerAgent generates its final answer for the first time. Stage 1: The ValidationAgent reorganizes the reasoning chain behind the ManagerAgent's answer, then extracts and prunes it. In this process, each reasoning step is explicitly labeled, providing structured input for next stage. Stage 2: For each step in the reasoning chain, we independently validate only the local inference between its premise and conclusion. For example, in $a \rightarrow b \rightarrow c$, we validate $a \rightarrow b$ and $b \rightarrow c$ separately. After obtaining all suggestions, they are fed back to the ManagerAgent as guidance for its subsequent actions (continue iterating or outputting the answer).

scattered execution steps into a logically coherent chain of reasoning. Each reasoning step is explicitly labeled and linked to its preceding steps through citations, ensuring that intermediate steps are properly referenced by subsequent steps, while the final step synthesizes all prior steps to derive a conclusion. The core objective of this stage is to transform scattered tool calls and observations into a clear, logically ordered reasoning chain, providing structured input for subsequent validation.

**Stage 2: Step-by-Step Reasoning validation.** In the second stage, the ValidationAgent independently verifies each reorganized reasoning step. For intermediate steps, the agent examines whether the inference logically follows from the referenced conditions by extracting these conditions, constructing *condition-inference* pairs, and applying a specialized validation template to evaluate correctness. For the final reasoning step, the agent not only checks logical soundness but also verifies whether the final answer properly addresses the original task. This process involves analyzing the task's implicit assumptions, confirming that the referenced conditions adequately satisfy these assumptions, validating the reasoning from conditions to the final answer, and ensuring that the answer fully adheres to the required format. For any problematic steps, the ValidationAgent provides concrete suggestions for improvement. If all steps are valid, it confirms the final answer as correct.

By isolating the reasoning process from its execution context, the logical correctness can be evaluated more objectively, while simultaneously enabling more precise localization of errors. By decomposing complex reasoning into atomic, independently verifiable steps, this method enhances validation precision and ensures errors remain fully traceable. The validation process is seamlessly integrated into the ManagerAgent's execution flow. If the answer passes validation, the ManagerAgent produces the final answer. Otherwise, the ManagerAgent re-evaluates the provided suggestions with full context and decides whether to overrule them or proceed with another round of iteration.

Table 1: Main experimental results. The best results are highlighted in **bold**, and the second-best are underlined. Results in gray are our own reproductions, as they were not officially reported. Except for methods marked with ‡, where the *Pass@n* metric was not explicitly stated in their official publications, all results are under the *Pass@1* metric. † denotes that the method is evaluated solely on the text-only subset of the GAIA validation set (103 samples).

| Method | Model | General AI Assistant (GAIA) | | | | WebWalkerQA |
|--------|-------|---------|---------|---------|------|-------------|
| | | Level 1 | Level 2 | Level 3 | Avg. | Avg. |
| *Closed-source Agent Systems* | | | | | | |
| OpenAI DR | O3 | 74.3 | 69.1 | 47.6 | 67.40 | - |
| *Open-sourced Agentic Systems With Mid-size LLMs* | | | | | | |
| Search-o1 | QwQ-32B-Base[1] | 53.8 | 34.6 | 16.6 | 39.80 | 34.1 |
| WebThinker | WebThinker-32B-RL | 56.4 | **50.0** | 16.7 | 48.50† | 46.50 |
| WebDancer | QwQ-32B-SFT | 46.1 | 44.2 | 8.3 | 40.70† | 38.40 |
| WebSailor | WebSailor-32B | - | - | - | 53.20† | - |
| WebShaper | QwQ-32B-SFT | **69.2** | **50.0** | 16.6 | **53.30**† | **49.70** |
| IAgent(Ours) | QwQ-32B-Base | 52.8 | 48.8 | **19.2** | 45.45 | 47.50 |
| *Open-sourced Agentic Systems With Large-size LLMs* | | | | | | |
| Agent-KB | GPT-4.1 | 79.25 | 58.14 | 34.62 | 61.21 | - |
| smolagents DR | GPT-4o | 67.92 | 53.49 | 34.62 | 55.15 | 46.50 |
| smolagents DR | GPT-4.1 | 69.81 | 60.47 | 34.62 | 59.39 | 53.00 |
| OWL | Claude-3.7-Sonnet | **84.91** | 68.60 | 42.31 | 69.70 | - |
| OAgents | Claude-3.7-Sonnet | 77.36 | 66.28 | 46.15 | 66.67 | - |
| WebExplorer‡ | Claude-4-Sonnet | - | - | - | 68.30† | 61.70 |
| BrowseMaster‡ | DeepSeek-R1-0528 | - | - | - | 68.00† | 62.10 |
| AWorld | Gemini-2.5-Pro, etc. | - | - | - | 67.89† | - |
| IAgent(Ours) | GPT-4o | 83.02 | 63.95 | 38.46 | 66.06 | 57.50 |
| IAgent(Ours) | GPT-4.1 | 83.02 | 67.44 | 38.46 | 67.88 | 59.00 |
| IAgent(Ours) | Claude-3.7-Sonnet | 84.62 | **72.41** | **50.00** | **72.73** | **68.50** |

# 4 EXPERIMENT

## 4.1 EXPERIMENTAL SETUP

**Benchmarks.** We evaluate our method on two challenging benchmarks: (i) GAIA (Mialon et al., 2023), a widely-adopted benchmark for General AI Assistants, with tasks spanning multimodal analysis, tool use, web search, and complex reasoning. This benchmark is organized into three difficulty levels from 1 (easiest) to 3 (hardest). (ii) WebWalkerQA (Wu et al., 2025b), which evaluates the ability of agents to browse subpages.

For GAIA, we use its entire public validation set, which consists of 165 queries. Some methods instead rely on the text-only subset of its validation set (103 samples, denoted by †), which is relatively simpler compared to the full validation set that includes multimodal queries. For WebWalkerQA, we randomly sample 200 examples. We adhere to the experimental setup of Webthinker (Li et al., 2025c), where the accuracy for these tasks is evaluated by Qwen2.5-72B-Instruct (Team, 2024).

**Models.** We design our framework based on smolagents (Roucher et al., 2025a), and conduct experiments across four model configurations. The first configuration uses QwQ-32B (Team, 2025), allowing comparison against agentic systems built on mid-size open-source LLMs. The remaining configurations employ GPT-4o (OpenAI, 2024), GPT-4.1 (OpenAI, 2025), and Claude 3.7 Sonnet (Anthropic, 2025b), respectively, enabling comparison against systems powered by large-size frontier LLMs.

---

[1]QwQ-32B-Base denotes the original QwQ-32B model without any fine-tuning.

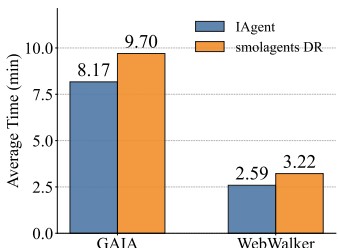

Figure 5: Average runtime per task.

Table 2: Ablation study of IAgent components. Results are reported on the GAIA benchmark.

| Method | Filter | Coder | Validation | Avg. |
|--------|--------|-------|------------|------|
| | ✓ | ✓ | ✓ | **62** |
| IAgent | ✗ | ✓ | ✓ | 58 |
| | ✓ | ✗ | ✓ | 58 |
| | ✓ | ✓ | ✗ | 56 |
| smolagents DR | – | – | – | 52 |

**Baselines.** We compare our approach with a proprietary agent as well as a broad set of leading open-source agentic systems. Specifically, we evaluate against the OpenAI DR agent (OpenAI, 2025) and two categories of open-source agents. The first category consists of open-source agentic systems built on mid-size open-source LLMs, primarily 32B-scale models. This group includes Search-o1 (Li et al., 2025b), WebThinker (Li et al., 2025c), WebDancer (Wu et al., 2025a), WebSailor (Li et al., 2025a), and WebShaper (Tao et al., 2025). The second category comprises open-source agentic systems powered by large-size LLMs, including smolagents DR (Roucher et al., 2025b), BrowseMaster (Pang et al., 2025), AWorld (Xie et al., 2025), OWL (Hu et al., 2025), OAgents (Zhu et al., 2025a), Agent-KB (Tang et al., 2025), and WebExplorer (Liu et al., 2025).

## 4.2 RESULTS

**Main Results.** Table 1 presents the main experimental results. Overall, IAgent consistently outperforms existing workflow baselines and achieves competitive or even superior results compared to both open-source and proprietary agents. Specifically: (i) In the comparison based on mid-size models, It consistently outperforms Search-o1, which also utilizes a non-fine-tuned base model. For the GAIA benchmark, IAgent maintains high performance on the more difficult Level 2 and Level 3 tasks, a success largely attributable to the enhanced reasoning and execution capabilities provided by the ValidationAgent and CoderAgent. On the WebWalkerQA benchmark, which focuses on retrieval within specific websites, the limited context window presents a challenge unique to mid-size models. Here, our filter module proves critical. By filtering out irrelevant webpages at an early stage, it drastically minimizes the accumulation of noise in the context, ensuring that the model's reasoning remains focused and effective despite the window constraints. (ii) The comparisons using GPT-4.1 and GPT-4o provide direct evidence of our framework's efficacy. Under both model configurations, IAgent achieves substantial improvements over the smolagents DR baseline on both GAIA and WebWalkerQA benchmarks, with gains reaching approximately 8 to 11 percentage points. These consistent margins demonstrate that the performance enhancements stem from our architectural innovations rather than solely the backbone model capabilities. (iii) When powered by Claude-3.7-Sonnet, IAgent establishes a new state-of-the-art on both benchmarks, achieving an average score of 72.73% on GAIA and 68.50% on WebWalkerQA. These results significantly outperform other systems utilizing the same backbone. Furthermore, IAgent even surpasses frameworks powered by more recent frontier models and the closed-source OpenAI DR system. (iv) As shown in Figure 5, the average runtime of IAgent is lower than that of smolagents DR. Furthermore, with the increase in task difficulty, the runtime savings achieved by IAgent become more pronounced.

**Ablation Study.** We conduct an ablation study to assess the effectiveness of IAgent's modular design. Specifically, we evaluate three ablated variants: (i) removing context-isolated filter module (Filter); (ii) removing CoderAgent (Coder); (iii) removing isolation-based stepwise validation module, which we implement as a ValidationAgent (Validation). All these experiments are conducted on 50 randomly selected queries from the GAIA benchmark with GPT-4o as the base model. The results are shown in Table 2, which demonstrate that the full IAgent design enables clear progression from novice to expert behavior. The removal of any single component results in an accuracy drop of at least 4%, validating the necessity of each module. Notably, even IAgent's ablated variants still outperform smolagents DR.

Table 3: Token Usage Comparison. Values represent the average token consumption per query on the GAIA benchmark.

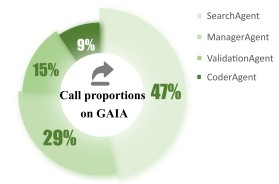

Figure 6: Agent distribution

| Method | Input Tokens | Output Tokens | Total Tokens |
|---|---|---|---|
| smolagents DR | 217.6k | 2.2k | 219.8k |
| IAgent | 144.1k ↓33.8% | 3.2k ↑45.5% | 147.3k ↓33.0% |

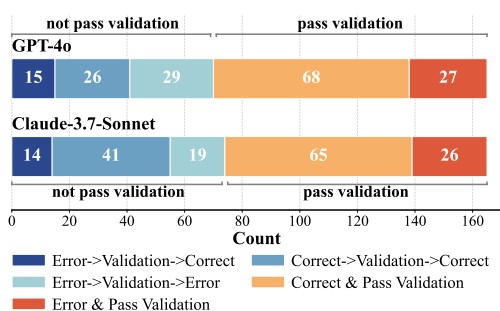

Figure 7: The effect of the ValidationAgent on the answers.

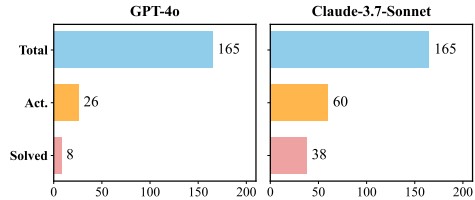

Figure 8: Activation details of CoderAgent. On GAIA, a total of 165 queries are evaluated. *Act.* denotes the number of queries where CoderAgent was invoked, and *Solved* denotes the number of queries correctly answered with its assistance.

### 4.3 FURTHER ANALYSIS

**IAgent significantly reduces the cost of answering.** We conducted a separate evaluation on the GAIA subset mentioned in section 4.2, with GPT-4o uniformly employed as the backbone model, and measured the computational cost of IAgent relative to the smolagents DR baseline, as shown in Table 3. On average, IAgent reduces the total token consumption per query by 33% compared to the smolagents DR baseline. Although IAgent's more responsive feedback mechanism results in an increase in output tokens per query, the overall monetary cost of completing a deep research task with IAgent remains substantially lower, as output tokens are negligible relative to input tokens. Figure 6 presents the utilization ratio of different agents in the GAIA benchmark, where SearchAgent constitutes the dominant share. The primary requirement for SearchAgent lies in long-context comprehension, an ability that most advanced LLMs can adeptly handle. We may therefore consider adopting more cost-effective models for this task to further reduce token consumption. For a detailed discussion on this aspect, refer to Appendix C.2.

**ValidationAgent and ManagerAgent complement in reasoning and evaluation.** Figure 7 illustrates the detailed effect of the ValidationAgent on the answers under two IAgent configurations in the GAIA benchmark. In both setups, at least 70 queries failed the initial validation after the first answer was generated. A portion of these were subsequently rectified upon receiving the ValidationAgent's feedback, thereby ultimately leading to the correct answer. In another set of instances, the ValidationAgent raised objections that the ManagerAgent, after re-evaluation, overruled, ultimately confirming the initial answer. These objections arose from two primary sources: either a premature conclusion by the ManagerAgent or the inherent limitations of the ValidationAgent's isolated, partial context. This dynamic highlights a complementary relationship: the ValidationAgent's context-isolated, stepwise validation and the ManagerAgent's holistic, context-aware reasoning work in concert to improve reliability. Furthermore, both configurations exhibited cases where the ValidationAgent failed to detect an incorrect final answer. This specific failure mode occurs when the SearchAgent reports factually erroneous information with high confidence. If the ManagerAgent's subsequent reasoning based on this erroneous premise is logically sound, the ValidationAgent, which primarily evaluates logical coherence, naturally cannot effectively identify such a fundamental error.

**Evaluations on BrowseComp.** We evaluate IAgent on a more challenging benchmark named BrowseComp (Wei et al., 2025), where single language models rarely answered correctly or scored.

Table 4: Results on BrowseComp Benchmark.

| Method & Model | Browsecomp |
|---|---|
| *LLM-based ReAct Agent* | |
| GPT-4.1 | 7.9% |
| Claude-3.7 | 4.8% |
| OpenAI-o1 | 14.3% |
| *Deep Research Agent* | |
| OAgents(Claude-3.7) | 22.2% |
| BrowseMaster(DeepSeek-R1) | **30.0%** |
| WebExplorer(Claude-4) | 12.2% |
| IAgent(Claude-3.7) | 26.0% |

Table 5: The effect of the context-isolated filter module on search efficiency. The table reports the average number of searches conducted by SearchAgent before and after the application of filtering. f/Ø denotes that no dedicated filter module is used, f/✓ denotes that the standard filter module is used, and f/□ denotes that the context-isolated filter module is used.

| Method | Before Filter | After Filter↓ |
|---|---|---|
| $IAgent^{f/Ø}$ | 9.8 | 9.8 |
| $IAgent^{f/✓}$ | 9.6 | 8.9 ↓9.18% |
| $IAgent^{f/□}$ | 9.4 | **7.6** ↓22.45% |

Given the scarcity of open-source frameworks evaluated on BrowseComp, we incorporated LLM-based ReAct Agents as additional baselines. As shown in Table 4, IAgent achieves exceptional performance on this benchmark, surpassing the Claude-4-Sonnet-powered WebExplorer. While IAgent trails the reasoning-model-driven BrowseMaster, this outcome is attributed to the high density of complex reasoning tasks within BrowseComp, which imposes significant demands on the reasoning capabilities of the foundation models. This observation is further corroborated by the ReAct Agent experiments, where the reasoning-enhanced OpenAI-o1 significantly outperforms general-purpose models such as GPT-4.1 and Claude-3.7-Sonnet.

**Key Component Analysis.** Figure 8 illustrates the significance of the CoderAgent. The IAgent tends to activate the CoderAgent when addressing complex problems. When activated, the success rate of solving these challenging tasks reaches 30.7% (8/26) or even higher. Meanwhile, we observe that different LLMs exhibit varying degrees of preference for activating the CoderAgent. Claude-3.7-Sonnet activates the CoderAgent more frequently compared to GPT-4o. Furthermore, due to Claude-3.7-Sonnet's superior coding capabilities, the CoderAgent driven by it achieves more outstanding performance in solving complex problems. For the filter module in SearchAgent, we adopt the GAIA subset mentioned earlier, which comprises 50 queries, and conducted a simple comparative experiment based on IAgent. With GPT-4o as the base model, the results in Table 5 show that our module effectively filters out irrelevant webpages. Finally, as shown in Figure 7, the Validation-Agent is able to correct at least 8.5% (14/165) of erroneous answers. Combined with the previous analysis, ValidationAgent and ManagerAgent jointly enhance the reliability of IAgent.

## 5 CONCLUSIONS

In this work, we propose IAgent, a web search agent framework that focuses on noise isolation and extended information access. By isolating the context in specific processes, IAgent effectively mitigates inertia bias and the sycophantic tendencies of LLMs, thereby enhancing the reliability of the final output. Furthermore, we introduce CoderAgent as a complement to web browsing, which leverages loop structures in code to automate large-scale data processing and further accesses structured data via unauthenticated APIs, significantly enhancing the efficiency of information acquisition. Our experiments demonstrate that IAgent not only achieves superior performance over both proprietary and open-source agents on challenging benchmarks but also reduces the total token cost by 33%.

## 6    THE USE OF LARGE LANGUAGE MODELS (LLMs)

We employed Large Language Models (LLMs) during the preparation of this paper. The main models we used were GPT-5 and Gemini-2.5-Pro. Their usage was limited to language-related assistance, including refining sentence clarity and style, generating tables for presentation purposes, and detecting formatting inconsistencies. All outputs produced by these models were carefully reviewed and verified by humans.

## 7    ETHICS STATEMENT

This work adheres to the ICLR Code of Ethics. All datasets and benchmarks used are publicly available, and we follow standard practices for their use. We believe our methods and results do not pose foreseeable risks of harm or misuse.

## 8    REPRODUCIBILITY STATEMENT

Our complete anonymized source code (including data processing, prompts, execution code, and evaluation scripts) is provided in the supplementary materials. For all models used in our experiments, we rely exclusively on their official APIs. The specific models are listed in Section 4, and the experimental settings as well as further implementation details are described in Appendix B.4.

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

## A  INERTIA BIAS

### A.1  THE PHENOMENON OF INERTIA BIAS IN THE SEARCH PROCESS

In our agent search pipeline, we observe a discrepancy between the ideal and the actual behavior of the LLM. Ideally, the model should stop the search process once it determines that all retrieved results are irrelevant. However, when the agent design incorporates the full history of previous steps (including the LLM's own queries), the model tends to continue selecting URLs even though it has already recognized them as irrelevant. This behavior reflects a form of path dependence or over-commitment.

- **Step 1: Query Generation** – The LLM produces a search query.

- **Step 2: Result Retrieval** – A search tool returns URLs and summaries.

- **Step 3 (Ideal)**: If all results are irrelevant, stop the search.

- **Observed Phenomenon**: When the agent includes its full history, the LLM often continues selecting URLs even though they appear irrelevant.

Figure 9 illustrates how IAgent mitigates the inertia bias during the search phase.

### A.2  QUANTITATIVE ANALYSIS OF INERTIA BIAS.

To provide a quantitative assessment of the impact of inertia bias on final performance, we conducted a rigorous manual statistical analysis of errors caused by search noise and contextual noise. We utilized the smolagents + GPT-4.1 setting on the GAIA benchmark.

Conducting this evaluation automatically is challenging due to the lack of specialized datasets for inertia bias and the inherent resilience of autonomous agents (where a failure in one iteration does not necessarily preclude success in the next). Therefore, we manually inspected the execution traces to ensure reliability, as using LLMs to statistically analyze these specific types of noise is currently neither realistic nor completely reliable.

**Impact of Search Noise.**  We analyzed the search noise resulting from inertia bias by recording the following metrics: (1) Total Searches: The total number of search actions performed. (2) Incorrect Paths due to Search Noise: The number of times search noise was not correctly identified, resulting in the agent following an incorrect path. (3) Total Sub-tasks: The total number of sub-tasks assigned to the SearchAgent (focusing on sub-tasks rather than macro-tasks, as each search is specific to a sub-task). (4) Sub-task Errors due to Search Noise: The number of times failure to identify search noise and following an incorrect path directly led to an erroneous conclusion for that sub-task. The results in Figure 10 indicate that search noise substantially impacts search efficiency, affecting 24.0% of total searches. Furthermore, although the agent can potentially compensate for invalid searches in subsequent iterations, the final conclusions of 13.5% of the sub-tasks are still adversely affected by this noise.

**Impact of Contextual Noise.**  We further analyzed errors caused by context bias (noise) by categorizing them into two distinct cases: Case 1: The context contained all observations necessary to derive the final answer, and these observations were correct; however, the ManagerAgent reached an incorrect conclusion due to interference from other irrelevant information. Case 2: The context contained only partial observations necessary for the final answer (which were correct); however, due to interference from other information, the ManagerAgent prematurely concluded that sufficient information had been obtained, leading to an erroneous conclusion. As shown in Figure 11, contextual noise significantly influences the final conclusion. Among the incorrect answers, a notable portion resulted specifically from the agent's inability to filter out contextual noise, either leading to misinterpretation of full information (Case 1) or premature termination based on partial information (Case 2).

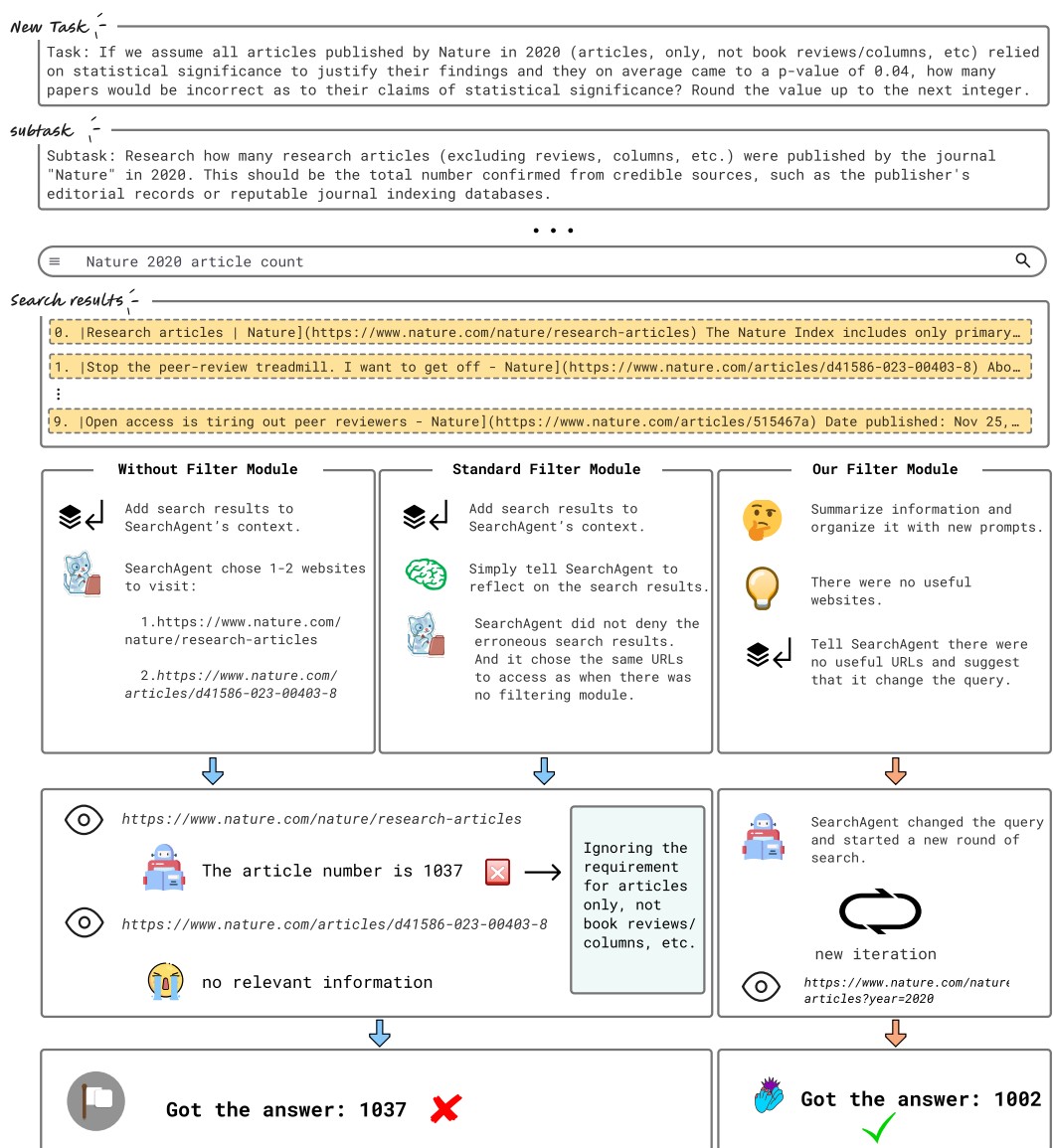

Figure 9: A comparison of search filter modules and their effect on answer accuracy. After obtaining the search results, the SearchAgent may process them in different ways. Without a filter module, the LLM simply selects a few seemingly most relevant webpages to visit. With a standard filter module, however, the selected pages are often still those that appear relevant but are in fact irrelevant, which is almost the same as the first case. In contrast, our filter module isolates the memory of the SearchAgent and retains only the most relevant content for evaluation, enabling a more objective judgment of relevance. It can also perform adaptive rewriting and re-iteration in time, thereby avoiding noise and making it easier to reach the correct answer.

## B DETAILS OF IAGENT

### B.1 DETAILS OF CODERAGENT

Algorithm 1 and 2 describes the complete data acquisition process of CoderAgent.

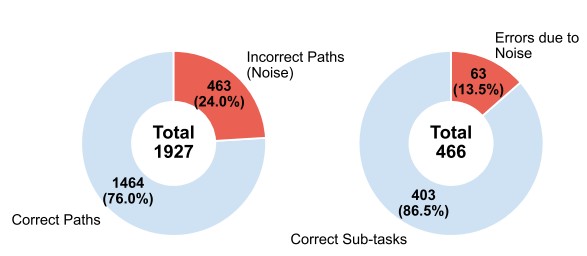

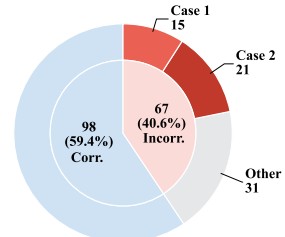

(a) Search Paths Analysis    (b) Sub-task Error Analysis

Figure 10: Analysis of search noise on GAIA. (a) Distribution of search paths, showing the proportion of incorrect paths caused by search noise. (b) Impact on sub-tasks, illustrating how search noise leads to erroneous conclusions.

Figure 11: Breakdown of errors by contextual noise on GAIA. The inner ring shows correct vs. incorrect answers, while the outer ring details the composition of errors: Case 1 (misinterpretation with complete observations) and Case 2 (premature termination with partial observations).

---

**Algorithm 1** CoderAgent: Intelligent Data Retrieval via API and Web Crawler

---

**Require:** Task description $T$, API Database $\mathcal{D}_{api}$, Environment Variables $\mathcal{E}$
**Ensure:** Final answer or data result $R$

1: **// Phase 1: Initialization**
2: $\mathcal{D}_{api} \leftarrow$ LoadAPIDatabases(MediaWiki, YouTube, ORCID)
3: $\mathcal{V} \leftarrow$ BuildTFIDFVectorizer($\mathcal{D}_{api}$)                    ▷ Build search index
4: $\mathcal{K} \leftarrow$ InjectAPIKeys($\mathcal{E}$)                    ▷ Inject YOUTUBE_API_KEY, etc.
5:
6: **// Phase 2: API Context Retrieval**
7: $\mathbf{q} \leftarrow$ TFIDFTransform($T, \mathcal{V}$)                    ▷ Vectorize task description
8: $\mathcal{S} \leftarrow$ CosineSimilarity($\mathbf{q}, \mathcal{D}_{api}$)                    ▷ Compute similarity scores
9: $\mathcal{E}_{top} \leftarrow$ TopK($\mathcal{S}, k = 3$)                    ▷ Retrieve top-$k$ relevant endpoints
10:
11: **// Phase 3: Generate Augmented Prompt**
12: $C_{api} \leftarrow$ GenerateAPIContext($\mathcal{E}_{top}$)                    ▷ Format endpoint documentation
13: $C_{auth} \leftarrow$ GetAuthRequirements($\mathcal{E}_{top}, \mathcal{K}$)                    ▷ Authentication hints
14: $P \leftarrow$ ConstructPrompt($T, C_{api}, C_{auth}$)                    ▷ Build LLM prompt
15:
16: **// Phase 4: Code Generation and Execution Loop**
17: **for** $i = 1$ **to** MaxSteps **do**
18:     thought, code $\leftarrow$ LLM($P$, history)                    ▷ Generate thought and code
19:
20:     **// Determine data retrieval method**
21:     **if** code contains API call from $\mathcal{E}_{top}$ **then**
22:         result $\leftarrow$ ExecuteAPIRequest(code, $\mathcal{K}$)                    ▷ Use structured API
23:     **else if** code contains web scraping logic **then**
24:         result $\leftarrow$ ExecuteWebCrawler(code)                    ▷ Fallback to crawler
25:     **else**
26:         result $\leftarrow$ ExecuteLocalComputation(code)                    ▷ Pure computation
27:     **end if**
28:
29:     observation $\leftarrow$ FormatOutput(result)
30:     history $\leftarrow$ history $\cup \{$(thought, code, observation)$\}$
31:     **if** IsFinalAnswer(code) **then**
32:         **return** result
33:     **end if**
34: **end for**
35: **return** TimeoutError

---

---

**Algorithm 2** LoadAPIDatabases: Build Structured API Knowledge Base

---

**Require:** API names list $\mathcal{A} = \{\text{MediaWiki}, \text{YouTube}, \text{ORCID}\}$
**Ensure:** API database $\mathcal{D}_{api}$
1: $\mathcal{D}_{api} \leftarrow \emptyset$
2: **for** each $a \in \mathcal{A}$ **do**
3:     json_data $\leftarrow$ LoadJSON($a$_api.json)
4:     **for** each endpoint $e \in$ json_data.endpoints **do**
5:         doc $\leftarrow$ $e$.name $\oplus$ $e$.action $\oplus$ $e$.description
6:         **for** each param $p \in e$.parameters **do**
7:             doc $\leftarrow$ doc $\oplus$ $p$.name $\oplus$ $p$.description
8:         **end for**
9:         $\mathcal{D}_{api} \leftarrow \mathcal{D}_{api} \cup \{(e, \text{doc})\}$
10:     **end for**
11: **end for**
12: **return** $\mathcal{D}_{api}$

---

### B.2 WEB BROWSER TOOLKIT

Table 6: Web browser toolkit

| Tool Name | Description | Input Parameters |
|---|---|---|
| web_search | Perform Google searches and retrieve search results | *query*, *filter_year (optional)* |
| fetch_html | Read and analyze the content of an HTML page | *url*, *query* |
| fetch_pdf | Read and analyze the content of a PDF page | *url*, *query* |
| visit_page | Returns transcript if the link is YouTube, otherwise downloads the web resource | *url* |
| find_archived_url | Finds historical content through web archive services | *url*, *date* |
| inspect_file_as_text | Reads the file as text and answers questions | *file_path*, *question (optional)* |

### B.3 HOLISTI-RAG HYBRID STRATEGY FOR THE WEB

HTML webpages and PDF files are the primary carriers of information in web search. We have optimized the processing of these resources and encapsulated them as tools for the SearchAgent. Previous methods also treated this process in isolation. However, the common method has been the pagination-based strategy, which simulates human browsing by analyzing content page by page (Roucher et al., 2025a; Tang et al., 2025; Zheng et al., 2025; Wu et al., 2025b). While intuitive, this method's computational cost grows quadratically with task complexity, as it must maintain the context of all previous interactions at each new step. In contrast, a holistic processing strategy analyzes the entire content of a webpage at once. For simple tasks requiring few interactions, the pagination-based method is more cost-effective. However, for complex deep research tasks that necessitate numerous interactions, the holistic strategy is markedly superior in cost-effectiveness and better preserves the content's semantic integrity. Given the rapid cost escalation of the pagination method for complex tasks, the holistic method often proves more advantageous in expectation.

Here is an example of web handling strategy costs. To formally analyze the trade-offs between pagination-based and holistic processing strategies, we model their respective computational costs.

The pagination-based strategy mimics human browsing, but its cost grows non-linearly. When processing the $i$-th interaction, the SearchAgent must maintain the complete context of the previous $i - 1$ interactions. For a task of complexity $C$ (defined as the number of interactions required), the total cost $L_p(C)$ is proportional to the sum of an arithmetic series. Ignoring cost from LLM response generation, the expected cost $E[L_p]$ is:

$$E[L_p] = \sum_{C=1}^{\infty} P(C) \cdot L_p(C) = \sum_{C=1}^{\infty} P(C) \left( n \frac{C(C+1)}{2} \right) \tag{1}$$

where $P(C)$ is the probability of a task having complexity $C$, and $n$ represents the average character count per page.

In contrast, the holistic strategy processes the entire content at once. Its expected cost $E[L_h]$ is constant, determined by the average content size of a webpage, $\bar{M}$:

$$E[L_h] = \sum_{C=1}^{\infty} P(C) \cdot \bar{M} = \bar{M} \tag{2}$$

Let $C_0$ be the cost equilibrium point where $L_p(C_0) = \bar{M}$. The expected cost difference $E[\Delta]$ between the two strategies is:

$$E[\Delta] = \sum_{C=C_0+1}^{\infty} P(C) \cdot \left(n\frac{C(C+1)}{2} - \bar{M}\right) - \sum_{C=1}^{C_0-1} P(C) \cdot \left(\bar{M} - n\frac{C(C+1)}{2}\right) \tag{3}$$

where the first term represents the expected cost savings of the holistic strategy for complex tasks ($C > C_0$), and the second term represents its expected loss for simpler tasks ($C < C_0$).

For medium-scale webpages (with character counts ranging from 50k to 200k, using 150k as the baseline), adopting a 10k character pagination strategy yields a cost balance point, namely $C_0 = 5$ when $L_p(C) = \bar{M}$. Based on the task difficulty distribution from the GAIA benchmark, we establish the following probability distribution:

- $P(C = 3) = 0.32$ (32% simple tasks)

- $P(C = 6) = 0.52$ (52% complex tasks)

- $P(C = 9) = 0.16$ (16% highly complex tasks)

The expected cost difference is given by:

$$E[\Delta] = P(C = 3) \cdot \Delta_3 + P(C = 6) \cdot \Delta_6 + P(C = 9) \cdot \Delta_9 \tag{4}$$

Substitution of the values shows that, under this task distribution, the overall processing strategy saves an average of 50,400 characters of token cost per page (approximately 33.6% of the additional cost). Moreover, in actual operation, the agent system is not limited to analyzing a single webpage; the number of interactions required by the aforementioned tasks will be greater, both of which further amplify the cost discrepancy.

Motivated by the above analysis, we designed two tools, `fetch_html` and `fetch_pdf`, and integrated them into the SearchAgent. These tools dynamically select the processing path based on a content length threshold. For content below this threshold, a holistic processing method is applied. For content exceeding it, the system switches to a standard RAG (Lewis et al., 2021)workflow, which first retrieves relevant text segments and then performs targeted information extraction. Equipped with these tools, the SearchAgent not only reduces overall cost but can also handle both exact and fuzzy matching problems. For complex queries requiring global ranking or aggregation (e.g., "the first paper published by this author"), it can also rely on the long-context integration and reasoning capabilities of the LLM.

## B.4 Implementation Details

On the GAIA and WebWalkerQA benchmarks, we set the temperature of all models to 0.0. For non-reasoning models, we limit the output token size to 4096.

In terms of tool integration, we have authorized multiple Python standard libraries and third-party libraries for model code execution. The model can dynamically load and invoke the following libraries, ensuring that IAgent is capable of handling complex input formats, including text, images, tables, and web data.

- requests
- zipfile
- os
- pandas
- numpy
- sympy

- json
- bs4
- pubchempy
- xml
- yahoo_finance
- Bio

- sklearn
- scipy
- pydub
- io
- PIL
- chess

- PyPDF2
- pptx
- torch
- datetime
- fractions
- csv

For all three benchmarks, we report only the *Pass@1* results, and all evaluations strictly follow the experimental protocols defined by GAIA and WebWalkerQA. To ensure objectivity, we restrict our comparison of baselines to single-attempt performance metrics. This includes explicitly reported *Pass@1* results and average accuracy scores from multiple experimental runs.

### B.5 A CASE OF A PIPELINE USING CODE TO REPRESENT ACTIONS

Figure 12 shows an example of a web search framework where actions are represented by code.

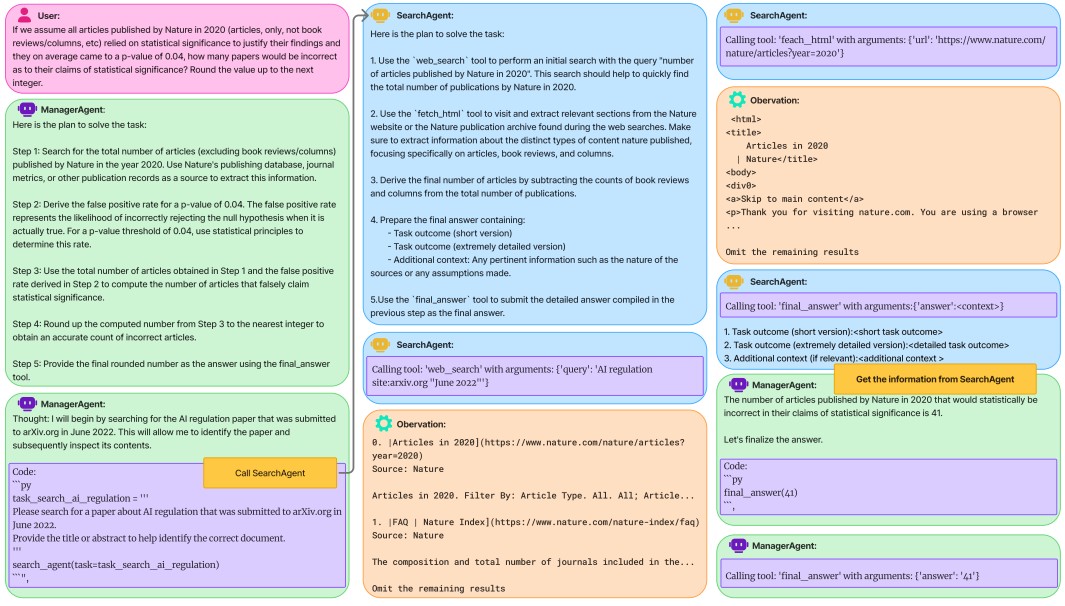

Figure 12: A web search framework that represents actions with code. After the user poses a question, the ManagerAgent invokes the SearchAgent through code, and the SearchAgent in turn invokes tools via code.

### B.6 DETAILS OF BENCHMARK EXPERIMENTS

During the evaluation of smolagents DR, we modified the original smolagents DR code and integrated (or ported) some of IAgent's tools into smolagents DR to ensure a fair comparison. For example, since the original smolagents tool for viewing XLSX files could not recognize colors, we redeveloped a tool capable of identifying both colors and background graphics, and implemented it concurrently in both IAgent and smolagents.

## C    TRY MORE MODEL CONFIGURATIONS

### C.1    COMBINE DEFFERENT LLMS TOGETHER

In the IAgent framework, assigning optimal models to agents according to their distinct responsibilities is essential for maximizing performance. In this section, we carefully select a set of hybrid models to demonstrate the potential of model composition. In this configuration, we employ Claude-4-Sonnet (Anthropic, 2025a) for the ManagerAgent, Gemini-2.5-Flash (DeepMind, 2025a) for the SearchAgent, and Gemini-2.5-Pro (DeepMind, 2025b) for the CoderAgent.

As exemplified by the activation details of the CoderAgent in Figure 8, which are obtained from the GAIA benchmark, the ManagerAgent demands exceptional instruction-following and task decomposition capabilities, which are notable strengths of Claude-4-Sonnet, leading to a higher activation frequency of the CoderAgent under this configuration. For the CoderAgent, Gemini-2.5-Pro surpasses GPT-4.1 in terms of reasoning, coding proficiency, and breadth of knowledge. This disparity explains why Config. 1 exhibits inferior performance relative to this configuration when the CoderAgent is activated. By contrast, the primary requirement for the SearchAgent is long-context comprehension, a capability that most advanced LLMs can adeptly handle. Consequently, we selected the more cost-effective Gemini-2.5-Flash model for this role.

By synthesizing the experimental results from Table 7 with the preceding analysis, we can demonstrate that a carefully curated ensemble of models yields superior performance at an equivalent cost when compared to the exclusive use of GPT-4o.

Table 7: Experimental results of IAgent with different model configurations.

| Method | Model | General AI Assistant (GAIA) | | | | WebWalkerQA |
|--------|-------|---------|---------|---------|------|------|
| | | Level 1 | Level 2 | Level 3 | Avg. | Avg. |
| Agent-KB | GPT-4.1 | 79.25 | 58.14 | 34.62 | 61.21 | - |
| smolagents DR | GPT-4o | 67.92 | 53.49 | 34.62 | 55.15 | 46.50 |
| OWL | Claude-3.7-Sonnet | 84.91 | 68.60 | 42.31 | 69.70 | - |
| OAgents | Claude-3.7-Sonnet | 77.36 | 66.28 | 46.15 | 66.67 | - |
| WebExplorer | Claude-4-Sonnet | - | - | - | 68.30 | 61.70 |
| BrowseMaster | DeepSeek-R1-0528 | - | - | - | 68.00 | 62.10 |
| AWorld | Gemini-2.5-Pro, etc. | - | - | - | 67.89 | - |
| IAgent | GPT-4o | 83.02 | 63.95 | 38.46 | 66.06 | 57.50 |
| IAgent | GPT-4.1 | 83.02 | 67.44 | 38.46 | 67.88 | 59.00 |
| IAgent | Claude-3.7-Sonnet | 84.62 | 72.41 | 50.00 | 72.73 | 68.50 |
| IAgent | Hybrid Models | 86.54 | 67.82 | 46.15 | 70.30 | 64.75 |

### C.2    COST ANALYSIS OF AGENT CONFIGURATIONS

To quantify the effect of model selection, we compute the expected cost per million tokens under the two configurations described in Section 4. We follow the official API prices, but distinguish between input- and output-based billing: for standard models (Gemini-2.5-Flash, Claude-4-Sonnet, GPT-4.1), we use their input token price; for reasoning-intensive models such as Gemini-2.5-Pro, we assume a balanced 1:1 ratio of input to output tokens, and thus use the average of its input ($1.25/M) and output ($10/M) costs, giving an effective price of $5.625/M tokens. This reflects the practical billing characteristics of mixed input-output usage.

Thus, it can be seen that the hybrid models deliver better performance while operating at a lower cost than the configuration that uses GPT-4o alone.

Table 8: Cost comparison between two agent configurations, per 1M tokens.

| Agent | Usage share | Hybrid Models | Price (USD/M) | Weighted cost |
|---|---|---|---|---|
| SearchAgent | 47% | Gemini-2.5-Flash | 0.30 (input) | 0.141 |
| ManagerAgent | 20% | Claude-4-Sonnet | 3.00 (input) | 0.600 |
| ValidationAgent | 15% | Gemini-2.5-Pro | 5.625 (avg.) | 0.844 |
| CoderAgent | 9% | Gemini-2.5-Pro | 5.625 (avg.) | 0.506 |
| **Total (GPT-4o-only)** | 100% | GPT-4o | 2.50 (input) | **2.500** |
| **Total (Hybrid Models)** | 100% | Mixed | – | **2.091** |

Table 9: Details on WebWalkerQA

| Method | Model | Easy | Medium | Hard | Average |
|---|---|---|---|---|---|
| *Open-source Agent Systems* | | | | | |
| WebThinker | WebThinker-32B-RL | 58.8 | 44.6 | 40.4 | 46.5 |
| WebDancer | QwQ-32B | 52.5 | 59.6 | 35.4 | 47.9 |
| WebShaper | Qwen-2.5-72B | 56.2 | 52.1 | 49.5 | 52.2 |
| BrowseMaster | DeepSeek-R1-0528 | - | - | - | 68.0 |
| WebExplore(scaffold) | Claude-4-Sonnet | - | - | - | 61.7 |
| *Ours* | | | | | |
| IAgent | Claude-4-Sonnet, etc. | 68.75 | 68.12 | 58.18 | 64.75 |

# D PROMPTS

In this section, we present some important prompts in IAgent.

## D.1 SYSTEM PROMPT FOR CODERAGENT

---

**System prompt for CoderAgent**

You are an expert assistant who can solve any task using code blobs. You will be given a task to solve as best you can.
To do so, you have been given access to a list of tools: these tools are basically Python functions which you can call with code.
To solve the task, you must plan forward to proceed in a series of steps, in a cycle of 'Thought:', 'Code:', and 'Observation:' sequences.

At each step, in the 'Thought:' sequence, you should first explain your reasoning towards solving the task and the tools that you want to use.
Then in the 'Code:' sequence, you should write the code in simple Python. The code sequence must end with '<end_code>' sequence.
During each intermediate step, you can use 'print()' to save whatever important information you will then need.
These print outputs will then appear in the 'Observation:' field, which will be available as input for the next step.
In the end you have to return a final answer using the `final_answer` tool.

Here are a few examples using notional tools:
---
(here omit the examples)

Here are the rules you should always follow to solve your task:
1. Always provide 'Thought:' and 'Code:' sections
2. Use authorized imports only: {{authorized_imports}}
3. Keep solutions simple and direct
4. Use `final_answer()` to return the final result

---

## D.2 PROMPT FOR CONTEXT-ISOLATED FILTER MODULE

---

**Prompt for context-isolated filtering module**

Here is the task:
```

 {{task}}
```

Here is the plan you once made:
```

 {{latest_plan}}
```

 Determine if the following websites can contribute to the task. If they can't, just reply "No". Otherwise, list 1-2 useful websites and their descriptions(including title, link, source and beginning words), the websites and their descriptions should be the same as the information I gave you.

 # Format 1 for Your Response
 "No"
 # Format 2 for Your Response
 <the website 1 contribute to the task>
 <the website 2 contribute to the task>(if there is website 2)
 (Tips:Do not provide any concluding statements at the beginning and end of the response)

 Here is the websites:
```

 {{raw_websites}}
```

---

## D.3 Prompt for isolated validation module-stage 1

---

**Prompt for isolated validation module-stage 1**

system: |-
 You are an expert at writing proofs based on the topic and the existing conditions. You write proofs in a clear and concise manner.

 Your every reasoning step will begin with `[reasoning n]:`, where n is the step number, starting from 1. Do not include the text [reasoning] anywhere else in the step. And in the end of every reasoning step, you sho use XML style tags `<ref>` to reference the step number or the reasoning step that you are referring to. If there are multiple references, you can use a semicolon to separate them, like `<ref>step 1;step 2</ref>`. If you refer to any prior step, use only step n, reasoning n, etc. without surrounding square brackets.

 Your task is to organize the reasoning process, ignore useless steps and analyze how to get the final answer. Do not verify the authenticity of each step and do not modify the answer.

 Output Your answer after the `Analysis:` section.
 Do not output anything except lines prefixed with `[reasoning n]:`.
 All reasoning steps [reasoning 1] to [reasoning n] must include a <ref> tag referencing previous steps. Steps [reasoning 1] to [reasoning n-1] must each be referenced at least once by later steps [reasoning 2] to [reasoning n], while [reasoning n] is the final conclusion and does not need to be referenced by any subsequent step.

  Here are a few examples of how to write the proof:
 ---
  (here omit the examples)

pre_messages: |-
 Your team has been working on a task an has got the final answer.
 But there may be some unuseful steps and wrong steps in the progress. Your task is to summarize the problem-solving process, ignore useless steps and summarize how your team came up with the final answer.

 You will be provided the task description, the final answer, and the steps your team has taken to solve the problem.

post_messages: |-
 Read the above conversation and output a Analysis to the question.

 Tips:
  1.not every step is useful, you should ignore the useless steps.
  2.The "Final answer" line is the target conclusion, not an evidence step; never reference it in <ref>. Your final [reasoning n] must synthesize the necessary prior reasoning steps to produce that answer.
  3.except for the last reasoning step, every intermediate reasoning step you write down must be used.

 Analysis:

---

## D.4 PROMPT FOR ISOLATED VALIDATION MODULE-STAGE 2 (INTERMEDIATE STEPS)

---

**Prompt for isolated validation module-stage 2**

check_intermediate_reasoning: |-
You are an expert who judges whether an inference is correct based on the existing information.

I will give you a series of basic conditions and a inference based on them, and you need to tell me if the inference is correct or not. These conditions are either the result of the execution of some tool or a sentence, and they are all absolutely correct.
You need to find any fallacies in your reasoning. You need to point out any unreasonable conclusions drawn from these conditions.

For each tool call, you should focus on whether the inference from Task outcome (extremely detailed version) to Task outcome (short version) is reasonable. If the detailed outcome lists multiple items for a category, the short outcome must not selectively omit any. Ensure No the summary's scope aligns with the detailed data's granularity.

Your response must be formatted using XML-style tags. Your response will be parsed automatically.
Your answer must strictly follow the three stages of `thought`, `judgment`, and `suggestions`. In the 'thought:' sequence, you should analyze the problem carefully, then put your analysis within <thought></thought> tags. In the `judgment` sequence, you should output your judgment. If the reasoning from the condition to the inference is correct, please output "True", otherwise output "False". Your judgment should be put within <judgment></judgment> tags. In the `suggestions` stage, if the condition is correct, output "No suggestions", otherwise output your guiding suggestions. Similarly, your suggestions should be put within <suggestions></suggestions> tags.

Here are a few examples of your response format:
---
(here omit the examples)

Now I will give you a series of conditions and an inference based on them, start you reasoning.

{{conditions}}

Inference: {{inference}}

your response:

---
(here omit the examples)

Now I will give you a series of conditions and an inference based on them, start you reasoning.

Task: {{task}}

{{conditions}}

Inference: {{inference}}

your response:

---

## D.5 PROMPT FOR ISOLATED VALIDATION MODULE-STAGE 2 (CONCLUSION STEP)

---

**Prompt for isolated validation module-stage 2**

check_final_reasoning: |-
  You are an expert who judges whether an inference is correct based on the existing information.

  I will give you a task, a series of basic conditions and a inference based on them, and you need to tell me if the inference is correct or not. These conditions are either the result of the execution of some tool or a sentence. And if they do not conflict with the conditions of the question, then they are absolutely correct. The inference is the final answer of the task. All conditions are set around the task, and you should carefully check whether the final reasoning meets the task requirements.
  You need to find any fallacies in your reasoning. You need to point out any unreasonable conclusions drawn from these conditions.
  Before answering, explicitly analyze the hidden assumptions of the question. You should check if the answer or conditions meets the hidden assumptions(e.g. a smooth surface implies that friction is neglected.). Distinguish between three types of failures: 1.Assumption Violation: The entity type does not possess this attribute. 2. Retrieval Failure: It should exist in theory, but cannot be found currently. 3. Non-existence: It truly does not exist or has not yet been assigned.
  Verify that the inference format does not contain 'none', 'N/A', '-', or any similar placeholder as an inference. When meet this situation, do not default to suggesting how to format this absence. Treat a "None" result as a red flag indicating a potential flaw in the reasoning process or the provided conditions. Your primary suggestion must be to challenge the steps that led to this null conclusion.
  For each tool call, you should focus on whether the inference from Task outcome (extremely detailed version) to Task outcome (short version) is reasonable. If the detailed outcome lists multiple items for a category, the short outcome must not selectively omit any. Ensure No the summary's scope aligns with the detailed data's granularity. Ensure no loss or unjustified compression of multi-item categories.
  When providing suggestions, your goal is to offer actionable, investigative steps to help the team solve the original task. Do not suggest changing the task itself or its output requirements. Instead, guide the team on how to work within the existing constraints to find a correct answer.

  You should follow the following steps to answer the question:
    1. Analyze the hidden assumptions of the question.
    2. Check if the conditions meets the hidden assumptions. If not, output "False" and provide a suggestion for team members to check conditions. In this case, your suggestion should not be to change the task. Instead, direct the team to Re-verify the source or Re-examine all conditions to check for any missing or potential answers. Your suggestions should be based on the hidden assumptions of the task (for example, if the hidden assumption is that the answer must be a place name, then you should suggest looking for the answer from the perspective of the place name).
    3. Check the reasoning from the known task and conditions to inferences as required.

  Your response must be formatted using XML-style tags. Your response will be parsed automatically.
  Your answer must strictly follow the three stages of `thought`, `judgment`, and `suggestions`. In the 'thought:' sequence, you should analyze the problem carefully, then put your analysis within <thought></thought> tags. In the `judgment` sequence, you should output your judgment. If the reasoning from the condition to the inference is correct, please output "True", otherwise output "False". Your judgment should be put within <judgment></judgment> tags. In the `suggestions` stage, if the condition is correct, output "No suggestions", otherwise output your guiding suggestions. Similarly, your suggestions should be put within <suggestions></suggestions> tags.

  Here are a few examples of your response format:
  ---
  (here omit the examples)

  Now I will give you a series of conditions and an inference based on them, start you reasoning.

  Task: {{task}}

  {{conditions}}

  Inference: {{inference}}

  your response:

---

