# OpenReview forum: "IAgent: A Web Search Framework for Noise Isolation and Extended Information Access"
_ICLR.cc/2026/Conference — ICLR 2026 Conference Withdrawn Submission_

### Official Review · Reviewer_FPft · 2025-10-23

**Soundness:** 3
**Presentation:** 3
**Contribution:** 3
**Rating:** 6
**Confidence:** 4

**Summary:**

The authors introduce IAgent, a novel multi-agent framework designed to address two critical challenges in existing systems: noise interference and monolithic information acquisition. IAgent has three main contributions: 1) A context-isolated filter module to mitigate search noise and LLM inertia bias at an early stage. 2) An isolation-based stepwise validation module to counteract contextual noise and sycophantic tendencies. 3) The introduction of a CoderAgent that extends information gathering capabilities beyond simple web browsing by leveraging APIs and programmatic automation. Experiments on the GAIA and WebWalkerQA datasets show that IAgent achieves better performance and reduces token costs by 38% compared to the baseline.

**Strengths:**

1. The design of IAgent appears to offer effective and insightful solutions to the drawbacks of current web search agents (noise interference and monolithic information acquisition).
2. The paper has Excellent Clarity and Presentation, and the figures illustrating the problems and methods are also clearly displayed.
3. According to the experimental results in the paper, IAgent outperforms the baselines on the two deep research benchmarks and consumes fewer tokens. Furthermore, the experiments are also quite comprehensive, including an effectiveness analysis for each module within IAgent.

**Weaknesses:**

1. When testing the cost of answering, the authors only experimented on the first 20 queries. This proportion is a bit small for a dataset with a total of 166 queries, and they were not a random sample of 20 queries. Whether the 38% token cost reduction can actually be achieved needs further verification.
2. In the experimental section, the backbone LLM used in IAgent and the various baselines are different, which may mean that the final experimental results are influenced by more than just the agent architecture.
3. The results in Table 1 have quite a few blanks. While this is certainly due to different agents testing on different benchmarks and the high cost of experimentation, if the results for WebWalkerQA could be completed for "Ours," it would help improve the robustness of the paper. Alternatively, use other datasets that smolagents has been tested on?

**Questions:**

1. Have the authors considered using a stronger open-source LLM, such as Qwen3-32B, as the backbone model to conduct unified testing across the different agent frameworks? This would help more rigorously test the performance differences between the various Agent architectures.

---

> ### Author Response · Authors · 2025-11-24
> **Part 1**
>
> Thank you for your insightful feedback. We have incorporated your suggestions into the revision to make paper more compelling. Below is a detailed response to each of your questions.
>
> **Weakness**
>
> **W1:** When testing the cost of answering, the authors only experimented on the first 20 queries. This proportion is a bit small for a dataset with a total of 166 queries, and they were not a random sample of 20 queries. Whether the 38% token cost reduction can actually be achieved needs further verification.
>
> **A1:** Your concern is reasonable, so we expanded the sample size to 50. Specifically, we randomly selected 50 queries from the GAIA benchmark and experimented with GPT-4o on both smolagents and IAgent. The experimental results are as follows:
>
> | Method        | Model  | Input Tokens             | Output Tokens        | Total Tokens             |
> | :------------ | :----- | :----------------------- | :------------------- | :----------------------- |
> | smolagents DR | GPT-4o | 217.6k                   | 2.2k                 | 219.8k                   |
> | IAgent        | GPT-4o | 144.1k $\downarrow$33.8% | 3.2k $\uparrow$45.5% | 147.3k $\downarrow$33.0% |
>
> The conclusion remains consistent with our previous findings: IAgent significantly reduces token consumption.
>
> **W2:** In the experimental section, the backbone LLM used in IAgent and the various baselines are different, which may mean that the final experimental results are influenced by more than just the agent architecture.
>
> **A2:** Thank you for your insightful suggestions. To ensure a more objective baseline comparison, we supplemented the following experiments: (i) Under the IAgent framework, we conducted experiments using GPT-4o and GPT-4.1 on the GAIA benchmark and the WebwalkerQA benchmark. (ii) Since the smolagents official report did not include experimental results using GPT-4o and GPT-4.1 on the GAIA benchmark, we ran the WebwalkerQA benchmark locally using GPT-4o and GPT-4.1. (iii) Under the IAgent framework, we conducted experiments using Claude-3.7-Sonnet on the GAIA benchmark and the WebwalkerQA benchmark. (iv) Furthermore, under the IAgent framework, we also conducted experiments using QWQ-32B on the GAIA benchmark and the WebwalkerQA benchmark.
>
> (i) and (ii) aim to perform a direct baseline comparison with smolagents using the same model (GPT-4o or GPT-4.1). (iii) is to demonstrate the superiority of IAgent over the baselines that employ some of the more cutting-edge models (e.g., OWL). (iv) is for comparison with frameworks using medium-sized models in Table 1 (e.g., WebThinker, WebDancer). Additionally, we have updated the evaluation method. Previously, we used character matching for evaluation, but now, following the approach of other baselines, we use an LLM for evaluation. The reselts are ase follows:
>
> | Method     | Model      | GAIA level 1 | GAIA level 2 | GAIA level 3 | GAIA avg. | WebWalkerQA |
> | ---------- | ---------- | ------------ | ------------ | ------------ | --------- | ----------- |
> | smolagents | GPT-4o     | 67.92        | 53.49        | 34.62        | 55.15     | 46.50       |
> | smolagents | GPT-4.1 | 69.81        | 60.47        | 34.62        | 59.39     | 53.0        |
> | IAgent     | QwQ-32B    | 52.8         | 48.8         | 19.2         | 45.45     | 47.50       |
> | IAgent     | GPT-4o     | 83.02        | 63.95        | 38.46        | 66.06     | 57.50       |
> | IAgent | GPT-4.1 | 83.02 | 67.44 | 38.46 | 67.88 | 59.0 |
> | IAgent     | Claude-3.7-Sonnet | 84.62        | 72.41        | 50.00        | 72.73     | 68.50       |
>
> We have updated Table 1. As the experimental results are numerous, we suggest you check our updated table for easy comparison.
>
> The experimental results show: 1. When using the same model (GPT-4o or GPT-4.1), IAgent shows an improvement of approximately 8-11 percentage points compared to smolagents. 2. When experimenting with Claude-3.7-Sonnet, IAgent still maintains a lead of at least 3 percentage points compared to other baselines. 3. Even when switching to a smaller model, IAgent can still achieve impressive performance, even exceeding some frameworks based on fine-tuned models.
>
> Finally, we realized that the discussion regarding combined models is an engineering-level issue. It is impossible to objectively compare the differences between frameworks. Therefore, we have placed the content regarding this aspect in Appendix C.

---

> ### Author Response · Authors · 2025-11-24
> **Part 2**
>
> **W3:** The results in Table 1 have quite a few blanks. While this is certainly due to different agents testing on different benchmarks and the high cost of experimentation, if the results for WebWalkerQA could be completed for "Ours," it would help improve the robustness of the paper. Alternatively, use other datasets that smolagents has been tested on?
>
> **A3:** Thank you for your suggestion. In the initial version, we were indeed constrained by the high cost of experimentation and did not fully complete a series of experiments on WebwalkerQA. However, it now seems that a small number of experimental results are not sufficiently convincing, so we have added the series of experiments mentioned in [W2]. We hope that the supplementary experimental results we have provided will be sufficiently persuasive.
>
> **Questions**
>
> **Q1:** Have the authors considered using a stronger open-source LLM, such as Qwen3-32B, as the backbone model to conduct unified testing across the different agent frameworks? This would help more rigorously test the performance differences between the various Agent architectures.
>
> **A1:** We are pleased to adopt your suggestion. As stated in [W2], we utilized QwQ-32B as the backbone model to conduct unified testing across the various agent frameworks. IAgent's performance not only surpasses frameworks also using QwQ-32B but even exceeds some frameworks based on fine-tuned models. This demonstrates that our framework generalizes well to mid-sized open-sourced models.

---

> > ### Comment · Reviewer_FPft · 2025-11-25
> >
> > I thank the authors for their response and additional experiments, which have partially addressed my concerns.

---

### Official Review · Reviewer_JkuJ · 2025-10-30

**Soundness:** 2
**Presentation:** 3
**Contribution:** 2
**Rating:** 2
**Confidence:** 3

**Summary:**

The authors introduce IAgent, an agent scaffold for web search tasks with specific filtering and context management steps for robustness and token efficiency. The authors argue that LMs exhibit “inertia bias” (i.e. they go along with their prior choices in context) and “contextual bias”, where increasing amounts of information and noise yield poor design choices by the agent. Their agent scaffold uses a “manager” and “worker/search” agent setup with a few extra design decisions: 1) to mitigate “inertia bias” due to noise, they include a filtering step of retrieved webpages for the “search” agent, 2) a module for reviewing the “manager” agent’s reasoning chain prior to producing a final answer, and 3) adding another worker agent with access to a code environment. They evaluate their approach on GAIA and WebWalker against other open-source (+ o3) approaches and provide a breakdown for how the agent solves these tasks.

**Strengths:**

1. IAgent improves over the general smolagent scaffold in both performance and cost on both GAIA and WebWalkerQA.
2. The general strategy of robust filtering of webpages and handling context management through separate LM calls is clever.
3. The analysis in Figure 6 and 7 is useful for understanding the reasoning trajectory of the agent with respect to each of its components.

**Weaknesses:**

1. Table 1 is confusing. There are several (10+) submissions on the public GAIA leaderboard [1, 2] with higher scores across all 3 levels, with many of these scaffolds being open source. It should be made more clear what the distinction between these results and the reported GAIA numbers are.
2. There are few ablations on the design of this scaffold, which includes several design choices on top of existing web agents. The primary analysis is Figure 6 and Figure 7, which show the agent usage / token breakdown of the main experiments, but it is unclear how to interpret the usefulness of each component separately.
3. Many of the baselines in Table 1 do not seem fair – the choice of model is not consistent across each agent scaffold, making it difficult to compare. The authors also argue that “carefully selecting model combinations is crucial”, but the baselines were not tuned fairly with this in mind.
4. There is little qualitative or quantitative analysis on where IAgent improves over existing, more task-specific scaffolds. smolagents is designed to be a general agentic scaffold, so the fact that IAgent can improve upon it is not very surprising.
5. In Table 5, we see that Config 2 uses Claude-4 and other newer frontier models. Many of the Table 1 experiments (including smolagent) do not use these newer models, which makes it unclear whether improvements are due to the scaffold or due to better base LMs.

**Questions:**

1. “Our experimental results demonstrate that IAgent achieves a significant performance enhancement over the smolagents baseline on the GAIA benchmark (Mialon et al., 2023) under similar model configurations.” Similar model configurations seems to just mean they use the same models, but IAgent appears to be a specific instantiation of the more general smolagents framework, so how much of its improvements over smolagent are attributed to more task-specific tuning for these types of tasks?
2. Why were smolagents results on WebWalkerQA not included? The paper states “As smolagents provide results solely on GAIA, we restrict evaluation of this configuration to the same benchmark”, but since the analysis centers on IAgent vs. smolagent, why not run smolagents on this benchmark?
3. Can you clarify how the results in Table 1 compare to the online reported results and why they were omitted / not included? Many officially reported scaffolds achieve 90%+ on GAIA Level 1 for example.

---

> ### Author Response · Authors · 2025-11-23
> **Part 1**
>
> We appreciate your insightful feedback. We have integrated your suggestions into the revision, which further enhances the paper's quality. Below is a detailed response to each of your questions.
>
> **Weakness:**
>
> **W1:** Table 1 is confusing. There are several (10+) submissions on the public GAIA leaderboard [1, 2] with higher scores across all 3 levels, with many of these scaffolds being open source. It should be made more clear what the distinction between these results and the reported GAIA numbers are.
>
> **A1:** We did not report model performance from the GAIA leaderboard for the following reasons: 1. The current results on the GAIA leaderboard are based on the GAIA benchmark test set, whereas almost all papers currently report results on the GAIA benchmark validation set. 2. There are frequent discrepancies in evaluation metrics across many results, particularly regarding the use of different Pass@K criteria. While some methods report Pass@1, others adopt more lenient metrics such as Pass@3 or even Pass@5. This inconsistency complicates fair comparisons between different agent frameworks and limits transparency regarding their practical capabilities. We only included frameworks whose accuracy rates are explicitly reported in official publications (articles or reports). Specifically, we focused on those providing clear pass@1 results, and frameworks explicitly stating pass@2 or higher were excluded. For frameworks that meet the first criterion (explicit accuracy reporting) but lack explicit statements regarding pass@1, we have added relevant notes in the paper. 3. We exhaustively collected all accessible experimental results from official publications prior to the ICLR deadline. At that time, IAgent (config 2) indeed achieved state-of-the-art performance among all open-source frameworks. However, we cannot guarantee that our collection covers every possible case, nor can we rule out the possibility that some studies may have updated their experimental data after the ICLR deadline.
>
> **W2:** There are few ablations on the design of this scaffold, which includes several design choices on top of existing web agents. The primary analysis is Figure 6 and Figure 7, which show the agent usage / token breakdown of the main experiments, but it is unclear how to interpret the usefulness of each component separately.
>
> **A2:** Thank you for highlighting this aspect. Figures 6 and 7 (renumbered as Figures 8 and 7 in the revised paper) were designed to analyze the influence of individual components on the final answer, thereby offering insights into their respective operational mechanisms. However, we acknowledge that these figures may not intuitively quantify the contribution of each component to the overall performance of IAgent. To address this, we have conducted an ablation study on IAgent to explicitly evaluate the impact of each component. We are pleased to inform you that this analysis has been incorporated into the revised paper (Table 2). The detailed results are as follows:
>
> | Method                                         | Avg  |
> | ---------------------------------------------- | ---- |
> | IAgent                                         | 62%  |
> | IAgent without context-isolation filter module | 58%  |
> | IAgent without CoderAgent                      | 58%  |
> | IAgent without ValidationAgent                 | 56%  |
> | smolagents                                     | 52%  |
>
> The removal of any single component results in an accuracy drop of at least 4%, validating the necessity of each module.

---

> ### Author Response · Authors · 2025-11-23
> **Part 2**
>
> **W3:** Many of the baselines in Table 1 do not seem fair – the choice of model is not consistent across each agent scaffold, making it difficult to compare. The authors also argue that "carefully selecting model combinations is crucial", but the baselines were not tuned fairly with this in mind.
>
> **A3:** Thank you for your insightful suggestions. To ensure a more objective baseline comparison, we supplemented the following experiments: (i) Under the IAgent framework, we conducted experiments using GPT-4o and GPT-4.1 on the GAIA benchmark and the WebwalkerQA benchmark. (ii) Since the smolagents official report did not include experimental results using GPT-4o and GPT-4.1 on the GAIA benchmark, we ran the WebwalkerQA benchmark locally using GPT-4o and GPT-4.1. (iii) Under the IAgent framework, we conducted experiments using Claude-3.7-Sonnet on the GAIA benchmark and the WebwalkerQA benchmark. (iv) Furthermore, under the IAgent framework, we also conducted experiments using QWQ-32B on the GAIA benchmark and the WebwalkerQA benchmark.
>
> (i) and (ii) aim to perform a direct baseline comparison with smolagents using the same model (GPT-4o or GPT-4.1). (iii) is to demonstrate the superiority of IAgent over the baselines that employ some of the more cutting-edge models (e.g., OWL). (iv) is for comparison with frameworks using medium-sized models in Table 1 (e.g., WebThinker). Additionally, we have updated the evaluation method. Previously, we used character matching for evaluation, but now, following the approach of other baselines, we use an LLM for evaluation. The reselts are ase follows:
>
> | Method     | Model      | GAIA level 1 | GAIA level 2 | GAIA level 3 | GAIA avg. | WebWalkerQA |
> | ---------- | ---------- | ------------ | ------------ | ------------ | --------- | ----------- |
> | smolagents | GPT-4o     | 67.92        | 53.49        | 34.62        | 55.15     | 46.50       |
> | smolagents | GPT-4.1 | 69.81        | 60.47        | 34.62        | 59.39     | 53.00        |
> | IAgent     | QwQ-32B    | 52.8         | 48.8         | 19.2         | 45.45     | 47.50       |
> | IAgent     | GPT-4o     | 83.02        | 63.95        | 38.46        | 66.06     | 57.50       |
> | IAgent | GPT-4.1 | 83.02 | 67.44 | 38.46 | 67.88 | 59.00 |
> | IAgent     | Claude-3.7-Sonnet | 84.62        | 72.41        | 50.00        | 72.73     | 68.50       |
>
> We have updated Table 1. As the experimental results are numerous, we suggest you check our updated table for easy comparison.
>
> The experimental results show: 1. When using the same model (GPT-4o or GPT-4.1), IAgent shows an improvement of approximately 8-11 percentage points compared to smolagents. 2. When experimenting with Claude-3.7-Sonnet, IAgent still maintains a lead of at least 3 percentage points compared to other baselines. 3. Even when switching to a smaller model, IAgent can still achieve impressive performance, even exceeding some frameworks based on fine-tuned models.
>
> **W4:** There is little qualitative or quantitative analysis on where IAgent improves over existing, more task-specific scaffolds. smolagents is designed to be a general agentic scaffold, so the fact that IAgent can improve upon it is not very surprising.
>
> **A4:** Although smolagents is a general-purpose agent framework, in a specific case within its repository, the smolagents team has also designed a specific workflow for deep research tasks, equipped with relevant tools (see: https://github.com/huggingface/smolagents/tree/main/examples/open_deep_research). We have made the following improvements to the framework for deep research tasks: we designed a filtering module to isolate two types of noise caused by inertia bias, which significantly impacts the correctness of deep research results. Simultaneously, we designed the CoderAgent, which utilizes APIs and code-based crawling to retrieve information, demonstrating the effectiveness of code-driven deep browsing in solving web search tasks that require multi-step iteration. We have supplemented this with ablation studies for each module, proving that these modules are effective for deep research tasks and can be transferred to other deep research frameworks.
>
> **W5:** In Table 5, we see that Config 2 uses Claude-4 and other newer frontier models. Many of the Table 1 experiments (including smolagent) do not use these newer models, which makes it unclear whether improvements are due to the scaffold or due to better base LMs.
>
> **A5:** Thank you for your feedback. As noted in [W3], we have supplemented a series of experiments to facilitate a relatively fair comparison with other frameworks. Regarding Config 2, our aim was to utilize a combination of more advanced models to balance accuracy and budget. However, we realized that the discussion regarding combined models is primarily an engineering-level issue, making it impossible to objectively compare the differences between frameworks. Therefore, we have moved the content regarding this aspect to Appendix B.

---

> ### Author Response · Authors · 2025-11-23
> **Part 3**
>
> **Questions:**
>
> **Q1** "Our experimental results demonstrate that IAgent achieves a significant performance enhancement over the smolagents baseline on the GAIA benchmark (Mialon et al., 2023) under similar model configurations." Similar model configurations seems to just mean they use the same models, but IAgent appears to be a specific instantiation of the more general smolagents framework, so how much of its improvements over smolagent are attributed to more task-specific tuning for these types of tasks?
>
> **A1:** As noted in [W4], smolagents has been specifically designed for deep research tasks, and all improvements in IAgent originate from the deep research example in the smolagents repository (https://github.com/huggingface/smolagents/tree/main/examples/open_deep_research). We have modified the phrasing in the paper to refer to this specific deep research instantiation as smolagents DR. The main improvements we made to it are also explained in [W4]. Additionally, the experimental setup for IAgent can be found in Appendix B.
>
> **Q2** Why were smolagents results on WebWalkerQA not included? The paper states "As smolagents provide results solely on GAIA, we restrict evaluation of this configuration to the same benchmark", but since the analysis centers on IAgent vs. smolagent, why not run smolagents on this benchmark?
>
> **A2:** Thank you for your helpful comments, this experiment deserves to be added to the article. We conducted the WebWalkerQA benchmark locally using GPT-4o, as stated in [W3]. The comparison with IAgent is presented below:
>
> | Method        | Model  | GAIA level 1 | GAIA level 2 | GAIA level 3 | GAIA avg. | WebWalkerQA |
> | :------------ | :----- | :----------- | :----------- | :----------- | :-------- | :---------- |
> | smolagents DR | GPT-4o | 67.92        | 53.49        | 34.62        | 55.15     | 46.50       |
> | IAgent        | GPT-4o | **83.02**    | **63.95**    | **38.46**    | **66.06** | **57.50**   |
>
> These experimental results demonstrate the superiority of IAgent over smolagents DR.
>
> **Q3** Can you clarify how the results in Table 1 compare to the online reported results and why they were omitted / not included? Many officially reported scaffolds achieve 90%+ on GAIA Level 1 for example.
>
> **A3:** We have already clarified this in [W1].
>
> ---
>
> We hope our response addresses your concern.

---

### Official Review · Reviewer_1Vm4 · 2025-10-30

**Soundness:** 2
**Presentation:** 3
**Contribution:** 2
**Rating:** 2
**Confidence:** 3

**Summary:**

The paper introduces context-isolated filtering module and isolation-based stepwise validation module for reducing noises occurred during search-intensive tasks. Additionally, the paper utilizes CodeAgent as an additional module for solving mathematical problems or repetitive subtasks. By integrating the proposed module on top of SmolAgent, IAgent improved peformances on the GAIA benchmark, and reduced the token cost.

**Strengths:**

* The paper points out concrete weaknesses in current web search agents (e.g., inertia bias) and frames “noise isolation & verification” as a main direction.
* The proposed modules are quite simple, well motivated, and empirically effective.
* The study provides clear quantitative insights into how each module contributes (e.g., validation correcting 6.7% of errors, filter reducing 27% redundant searches).

**Weaknesses:**

[W1] While the proposed modules are intuitive and simple, the proposed modules lack research novelty but rather a straightforward application aimed at deep research tasks.

[W2] Different backbone across methods
* In table 1, backbone LLMs for baselines are different from that of IAgent. As the main contribution of this work is orthogonal to choice of the backbone LLM, the authors should demonstrate superiority of IAgent over the baselines (e.g., SmolAgent, OWL) under identical backbone LLMs.

* In line 367: “The first configuration (Config. 1), intended to provide a direct baseline comparison with smolagents, powers all agents with GPT-4.1 (OpenAI, 2025), whose performance is similar to that of GPT-4o (OpenAI, 2024).”, why not compare IAgent and SmolAgent under GPT-4o? I think “performance of GPT-4o and GPT-4.1 is similar” is not a valid reason for comparing IAgent + GPT-4.1 with SmolAgent + GPT-4o.

If the authors show that IAgent is remarkably superior to SmolAgent under the same LLM backbone, I will raise my recommendation.

[W3] Why did the authors not provide results corresponding to SmolAgent in WebWalker? As SmolAgent is a direct baseline, I think the authors need to show the superiority of IAgent over SmolAgent in WebWalker benchmark.


[W4] Different subset sampling across analysis.

Analysis regarding token usage and ablations on “filter module in Search Agent” are done on subset of GAIA benchmark (20 tasks out of 165 tasks), but the subsets are not unified (prior one is first 20 problems, and second one is randomly sampled 20 problems). Authors should fix the subset, and conduct additional analysis on the fixed subset.

**Questions:**

[Q1] What is the performance of IAgent without CoderAgent?

[Q2] The tables are mainly presents final score on each benchmark. Is it possible to quantitively analyze whether the context-isolation module indeed reduces inertia bias?

---

> ### Author Response · Authors · 2025-11-23
> **Part 1**
>
> Thank you for your insightful feedback. We have incorporated your suggestions into the revision to make paper more compelling. Below is a detailed response to each of your questions.
>
> **Weakness**
>
> **W1:** While the proposed modules are intuitive and simple, the proposed modules lack research novelty but rather a straightforward application aimed at deep research tasks.
>
> **A1:** Our core contributions are two twofold: 1. We discovered the inertia bias problem of LLMs in agent applications, and identified the noise caused by inertia bias in deep research tasks and proposed an isolation-based solution. We believe inertia bias is a universal problem in agent applications. As LLM agents undertake more complex, context-rich tasks, inertia bias is unavoidable. For example, the problem that often occurs in the field of code generation where an agent makes a mistake in the initial debugging approach and then falls into an infinite loop (when the project is very complex), and we hope our method can provide a line of thinking for solving such problems. 2. The core contribution of CoderAgent is the specialized design of code-driven deep web browsing. Although previous agent frameworks have adopted code act as action representation, they did not specifically design the web browsing part for the deep research field. Our work proves the effectiveness of code-driven deep browsing in solving web search tasks that require multi-step iteration.
>
> **W2:** Different backbone across methods
>
> - In table 1, backbone LLMs for baselines are different from that of IAgent. As the main contribution of this work is orthogonal to choice of the backbone LLM, the authors should demonstrate superiority of IAgent over the baselines (e.g., SmolAgent, OWL) under identical backbone LLMs.
> - In line 367: “The first configuration (Config. 1), intended to provide a direct baseline comparison with smolagents, powers all agents with GPT-4.1 (OpenAI, 2025), whose performance is similar to that of GPT-4o (OpenAI, 2024).”, why not compare IAgent and SmolAgent under GPT-4o? I think “performance of GPT-4o and GPT-4.1 is similar” is not a valid reason for comparing IAgent + GPT-4.1 with SmolAgent + GPT-4o.
>
> If the authors show that IAgent is remarkably superior to SmolAgent under the same LLM backbone, I will raise my recommendation.
>
> **A2:** Thank you for your insightful suggestions. To ensure a more objective baseline comparison, we supplemented the following experiments: (i). Under the IAgent framework, we conducted experiments using GPT-4o on the GAIA benchmark and the WebwalkerQA benchmark. (ii). Since the smolagents official report did not include experimental results using GPT-4o on the GAIA benchmark, we ran the WebwalkerQA benchmark locally using GPT-4o. (iii). Under the IAgent framework, we conducted experiments using Claude-3.7 on both benchmarks. (iv). Furthermore, under the IAgent framework, we also conducted experiments using QWQ-32B  on both benchmarks.
>
> (i) and (ii) aim to perform a direct baseline comparison with smolagents using the same model (GPT-4o). (iii) is to demonstrate the superiority of IAgent over the baselines that employ some of the more cutting-edge models (e.g., OWL). (iv) is for comparison with frameworks using medium-sized models in Table 1 (e.g., WebThinker, WebDancer). Additionally, we have updated the evaluation method. Previously, we used character matching for evaluation, but now, following the approach of other baselines, we use an LLM for evaluation. The reselts are ase follows:
>
> | Method        | Model             | GAIA level 1 | GAIA level 2 | GAIA level 3 | GAIA avg. | WebWalkerQA |
> | ------------- | ----------------- | ------------ | ------------ | ------------ | --------- | ----------- |
> | smolagents  | GPT-4o            | 67.92        | 53.49        | 34.62        | 55.15     | 46.50       |
> | IAgent | QwQ-32B      | 52.8         | 48.8         | 19.2         | 45.45     | 47.50       |
> | IAgent  | GPT-4o            | 83.02        | 63.95        | 38.46        | 66.06     | 57.50       |
> | IAgent | Claude-3.7-Sonnet | 84.62        | 72.41        | 50.00        | 72.73     | 68.50       |
>
> We have updated Table 1. As the experimental results are numerous, we suggest you check our updated table for easy comparison.
>
> The experimental results show: 1. When using the same model (GPT-4o), IAgent shows an improvement of approximately 11 percentage points compared to smolagents. 2. When experimenting with Claude-3.7-Sonnet, IAgent still maintains a lead of at least 3 percentage points compared to other baselines. 3. Even when switching to a smaller model, IAgent can still achieve impressive performance, even exceeding some frameworks based on fine-tuned models.
>
> Finally, we realized that the discussion regarding combined models is an engineering-level issue. It is impossible to objectively compare the differences between frameworks. Therefore, we have placed the content regarding this aspect in Appendix C.

---

> ### Author Response · Authors · 2025-11-23
> **Part 2**
>
> **W3:** Why did the authors not provide results corresponding to SmolAgent in WebWalker? As SmolAgent is a direct baseline, I think the authors need to show the superiority of IAgent over SmolAgent in WebWalker benchmark.
>
> **A3:** As noted in [W2], since smolagents has not officially reported results using GPT-4o on the WebwalkerQA benchmark, we conducted corresponding experiments to demonstrate effectiveness.
>
> **W4:** Different subset sampling across analysis. Analysis regarding token usage and ablations on “filter module in Search Agent” are done on subset of GAIA benchmark (20 tasks out of 165 tasks), but the subsets are not unified (prior one is first 20 problems, and second one is randomly sampled 20 problems). Authors should fix the subset, and conduct additional analysis on the fixed subset.
>
> **A4:** We apologize for the misunderstanding. In fact, both the Analysis regarding token usage and the ablations on the "filter module in Search Agent" were conducted on the first 20 problems of the GAIA benchmark. Since the problems in the GAIA benchmark itself are randomly ordered (levels 1-3 are randomly distributed), we initially used the term "random" in the paper. We have corrected the expression in the revised paper. Furthermore, we expanded the scope of the Analysis regarding token usage experiment to 50 problems to ensure the objectivity and fairness of our analysis. The supplementary experiments indicate that IAgent can reduce token consumption by 33% (Table 3) and is able to filter 23% of irrelevant webpages (Table 5).
>
> **Questions:**
>
> **Q1** What is the performance of IAgent without CoderAgent?
>
> **A1:** Thank you for recommending an ablation study on IAgent to evaluate the impact of CoderAgent. We are pleased to inform you that this analysis has been included in the revised paper (Table 2). The detailed results are as follows:
>
> | Method                                         | Avg  |
> | ---------------------------------------------- | ---- |
> | IAgent                                         | 62%  |
> | IAgent without context-isolation filter module | 58%  |
> | IAgent without CoderAgent                      | 58%  |
> | IAgent without ValidationAgent                 | 56%  |
> | smolagents                                     | 52%  |
>
> When not using the Coder component, the performance of IAgent drops by 4 percentage points.
>
> **Q2** The tables are mainly presents final score on each benchmark. Is it possible to quantitively analyze whether the context-isolation module indeed reduces inertia bias?
>
> **A2:** The ablation study in [Q1] also includes the performance of IAgent without the context-isolation filter module. The results show that when the context-isolation module is not used, the performance of IAgent drops by 4 percentage points.

---

> ### Comment · Reviewer_1Vm4 · 2025-11-24
>
> Thank you for the response. Here are a few comments regarding your response.
>
> ---
>
> **Regarding A2**
>
> As far as I remember, in original manuscript, there was a comparison between smolagent + GPT-4o vs IAgent + GPT-4.1.
> In current manuscript, IAgent + GPT-4o is only compared against SmolAgent + GPT-4o.
> What is the performance improvement IAgent + GPT-4.1 over SmolAgent + GPT-4.1?
>
> ---
>
> **Regarding A1**
>
> 1. I like the framing of an inertia bias problem, but it would be better if you provide more thorough analysis regarding the problem in your manuscript. Although the ablation studies done during the rebuttal period support the significance of the inertia bias, It would be better if you provide analysis like: Presenting fraction of the tasks in GAIA that are failed by Baseline Agent (e.g., SmolAgent) due to the inertia bias, and how much of them are resolved by IAgent framework.
>
> 2. Although you mention that employing CodeAct mechanism to deep web browsing adds on novelty, but I respectfully disagree with your claim. I believe an employment of CodeAct to specific domain (i.e., deep research) does not include research novelty, rather it is a application / extension of CodeAct. Additionally, it is quite irrelevant to the inertia bias that the authors initially framed.

---

> ### Author Response · Authors · 2025-11-27
> **R2-Part 1**
>
> Thank you for your response and the follow-up questions!
>
> **R2-Q1:** Regarding A2: As far as I remember, in original manuscript, there was a comparison between smolagent + GPT-4o vs IAgent + GPT-4.1. In current manuscript, IAgent + GPT-4o is only compared against SmolAgent + GPT-4o. What is the performance improvement IAgent + GPT-4.1 over SmolAgent + GPT-4.1?
>
> **R2-A1:** Thank you for your valuable question. We have added experiments to compare IAgent + GPT-4.1 and smolagents + GPT-4.1 on both benchmarks. Inspired by Reviewer cWmy, we identified an issue with the initial experimental setup for IAgent and consequently re-conducted the experiments for IAgent + GPT-4.1 on GAIA. When utilizing GPT-4.1, both smolagents and IAgent achieved slight performance improvements. However, IAgent continues to demonstrate a significant advantage over smolagents. We have incorporated these experimental results into the paper (Table 1).
>
> | Method     | Model   | GAIA level 1 | GAIA level 2 | GAIA level 3 | GAIA avg. | WebWalkerQA |
> | ---------- | ------- | ------------ | ------------ | ------------ | --------- | ----------- |
> | smolagents | GPT-4.1 | 69.81        | 60.47        | 34.62        | 59.39     | 53.0        |
> | IAgent     | GPT-4.1 | **83.02**    | **67.44**    | **38.46**    | **67.88** | **59.0**    |

---

> > ### Author Response · Authors · 2025-11-27
> > **R2-Part 2**
> >
> > **R2-Q2:** Regarding A1: I like the framing of an inertia bias problem. However, authors did not proposed any metrics or detailed ablations (e.g., how much fractions of tasks are failed due to inertia bias).
> >
> > **R2-A2:** We are pleased to discuss this issue with you. Inertia bias was a phenomenon we serendipitously observed during our experiments. Initially, we intended to evaluate it quantitatively. However, there currently appears to be no dataset specifically designed for this problem (e.g., a dataset that provides a research topic, a series of queries, and corresponding web search results, requiring an LLM to determine whether these search results genuinely contribute to the final answer).
> >
> > Furthermore, evaluating this with the deep research benchmarks is also extremely challenging. Since the subject is an autonomous agent, a failure in a previous search and web browsing session does not preclude the possibility of finding the correct path in subsequent iterations. Consequently, it is difficult to quantitatively determine solely from the final outcomes how many task failures were directly caused by search noise resulting from inertia bias. Nevertheless, the wasted steps and erroneous answers potentially caused by inertia bias during this process are objectively real. Statistical analysis of context bias resulting from inertia bias is also an exceedingly tedious process. It requires inspecting every observation preceding the conclusion to verify its correctness and contribution to the final answer, and ultimately determining whether the final error resulted from context bias. It is fair to say that using LLMs to statistically analyze the aforementioned two types of noise in deep research is neither realistic nor reliable. To provide a quantitative analysis of the impact of inertia bias on the final results, we conducted a manual statistical analysis of errors caused by search noise and contextual noise for smolagents + GPT-4.1 on the GAIA benchmark, despite the significant time cost involved. Specifically:
> >
> > ***(a) Search noise caused by inertia bias.***
> > We recorded: 1. The total number of searches. 2. The number of times search noise was not correctly identified, resulting in following an incorrect path. 3. The total number of sub-tasks assigned to the SearchAgent (we focus on sub-tasks rather than macro-tasks here, as each search by the SearchAgent is conducted specifically for a sub-task). 4. The number of times failure to identify search noise and following an incorrect path led to an erroneous conclusion for that sub-task. The results are as follows:
> >
> > | Total Searches | Incorrect Paths due to Search Noise | Total Sub-tasks | Sub-task Errors due to Search Noise |
> > | :------------- | :---------------------------------- | :-------------- | :---------------------------------- |
> > | 1927           | 463                                 | 466             | 63                                  |
> >
> > These results indicate that search noise substantially impacts search efficiency, affecting 24.0% (463/1927) of searches. Furthermore, although the agent can potentially compensate for invalid searches in subsequent iterations, the final conclusions of 13.5% (63/466) of the sub-tasks are still adversely affected.
> >
> > ***(b) Errors caused by contextual noise.***
> > We categorized these errors into two cases:  Case 1: The context contained all observations necessary to derive the final answer, and these observations were correct, but the ManagerAgent reached an incorrect conclusion due to interference from other information. Case 2: The context contained only partial observations necessary for the final answer (which were correct), but due to interference from other information, the ManagerAgent prematurely concluded that sufficient information had been obtained, leading to an erroneous conclusion. The results are as follows:
> >
> > | Total Questions | Incorrect Answers | Errors due to contextual noise (Case 1) | Errors due to contextual noise (Case 2) |
> > | :-------------- | :---------------- | :-------------------------------------- | :-------------------------------------- |
> > | 165             | 67                | 15                                      | 21                                      |
> >
> > This similarly demonstrates that contextual noise significantly influences the final answer.
> >
> > We have added this detailed analysis to Appendix A.2 to provide stronger evidence for our claim.

---

> > > ### Author Response · Authors · 2025-11-27
> > > **R2-Part 3**
> > >
> > > **R2-Q3:** Regarding A: Although you mention that employing CodeAct mechanism to deep web browsing adds on novelty, but I respectfully disagree with your claim. I believe an employment of CodeAct to specific domain (i.e., deep research) does not include research novelty, rather it is a application / extension of CodeAct.
> > >
> > > **R2-A3:** First, we want to clarify that the primary innovation of the CoderAgent lies in leveraging CodeAct to guide the LLM in utilizing APIs or web crawlers to retrieve batch information or content protected by JavaScript, along with configuring a dedicated experimental environment for it, rather than claiming to be the first to apply CodeAct in the deep research domain generally. We acknowledge that this constitutes an application or extension of CodeAct. Consequently, our Method section primarily focuses on inertia bias and the corresponding proposed solutions, rather than centering solely on the CoderAgent. We included the description of this component in the paper for two reasons: 1. To the best of our knowledge, no such agent has been previously designed within the deep research domain, omitting it would potentially prevent readers from fully comprehending our framework's architecture.  2. It is an integral part of our framework that significantly enhances performance.
> > >
> > > ---
> > >
> > > We hope these explanations address your inquiries. If you have any more questions, please let us know. We're always happy to answer them!

---

> > > ### Comment · Reviewer_1Vm4 · 2025-11-27
> > >
> > > Thank you for the clarification. Although my initial concerns regarding fair comparison with baselines are addressed, I am not convinced with the novelty of this paper. Hence, I changed my recommendation to 4.

---

### Official Review · Reviewer_cWmy · 2025-11-03

**Soundness:** 3
**Presentation:** 3
**Contribution:** 3
**Rating:** 4
**Confidence:** 4

**Summary:**

The paper introduces IAgent, an agentic web search framework that explicitly addresses the limitations of Deep Research methods corresponding to noisy search results and incorrect reasoning steps. Specifically, a filtering module is introduced to remove irrelevant webpages  from the search results. Next, a reasoning verification module is added to validate the correctness of the reasoning process in a stepwise fashion. The paper also adds a code-execution approach when multiple parallel iterations of search calls are necessary. The method is benchmarked on popular evaluation sets, such as GAIA and WebwalkerQA.

**Strengths:**

1. The methodology is described very well, with clear figures and the paper is easy to read.
2. The experimental results are impressive, with considerably better performance than the Smolagents baseline

**Weaknesses:**

1. The baseline comparison is not apples-to-apples since a variety of models are used, making it hard to understand how reliable the improvements are. The smolagents baselines uses GPT-4o while IAgent approach uses considerably more recent/performant models. The authors should consider adding a GPT-4.1 variant of Smolagents.
2. While there is an overall improvement over Smalagents, there is a significant degradation in performance on level-3, representing the more difficult subset. This brings into question whether the paper’s method is mainly effective for the more easier deep research questions. I would have expected the opposite, since the proposed CoderAgent and reasoning-verification should be more effective for the harder tasks. The authors should consider benchmarking on BrowseComp, which has been considered much more harder benchmark for deep research.
3. The approach in the paper looks highly hand-stitched and only relying on closed-source models, making it unclear how to pursue further improvements with training. The authors should consider showing performance with an open-source model like GPT-OSS-20B or Qwen3-32B. Would be great if they can further improve performance of these models with an approach like SFT over closed-source model trajectories.
4. The related work section needs more revision, specifically to incorporate discussion on how the proposed method differs from existing multi-agent methods. The related work currently just reads like a summary of prior work in each sub-category.

**Questions:**

1. While token-count cost is compared, how does the comparison of inference time look like for IAgent vs Smolagents when using the same LLM?

---

> ### Author Response · Authors · 2025-11-23
> **Part 1**
>
> Thank you for your valuable and constructive review.  We have integrated your suggestions into the revision, which further enhances the paper's quality. Below is a detailed response to each of your questions.
>
> **Weakness**
>
> **W1:** The baseline comparison is not apples-to-apples since a variety of models are used, making it hard to understand how reliable the improvements are. The smolagents baselines uses GPT-4o while IAgent approach uses considerably more recent/performant models. The authors should consider adding a GPT-4.1 variant of Smolagents.
>
> **A1:** Thank you for your insightful suggestions. To ensure a more objective baseline comparison, we supplemented the following experiments: (i) Under the IAgent framework, we conducted experiments using GPT-4o and GPT-4.1 on the GAIA benchmark and the WebwalkerQA benchmark. (ii) Since the smolagents official report did not include experimental results using GPT-4o and GPT-4.1 on the GAIA benchmark, we ran the WebwalkerQA benchmark locally using GPT-4o and GPT-4.1. (iii) Under the IAgent framework, we conducted experiments using Claude-3.7-Sonnet on the GAIA benchmark and the WebwalkerQA benchmark. (iv) Furthermore, under the IAgent framework, we also conducted experiments using QWQ-32B on the GAIA benchmark and the WebwalkerQA benchmark.
>
> (i) and (ii) aim to perform a direct baseline comparison with smolagents using the same model (GPT-4o or GPT-4.1). (iii) is to demonstrate the superiority of IAgent over the baselines that employ some of the more cutting-edge models (e.g., OWL). (iv) is for comparison with frameworks using medium-sized models in Table 1 (e.g., WebThinker, WebDancer). Additionally, we have updated the evaluation method. Previously, we used character matching for evaluation, but now, following the approach of other baselines, we use an LLM for evaluation. The reselts are ase follows:
>
> | Method     | Model      | GAIA level 1 | GAIA level 2 | GAIA level 3 | GAIA avg. | WebWalkerQA |
> | ---------- | ---------- | ------------ | ------------ | ------------ | --------- | ----------- |
> | smolagents | GPT-4o     | 67.92        | 53.49        | 34.62        | 55.15     | 46.50       |
> | smolagents | GPT-4.1 | 69.81        | 60.47        | 34.62        | 59.39     | 53.00        |
> | IAgent     | QwQ-32B    | 52.8         | 48.8         | 19.2         | 45.45     | 47.50       |
> | IAgent     | GPT-4o     | 83.02        | 63.95        | 38.46        | 66.06     | 57.50       |
> | IAgent | GPT-4.1 | 83.02 | 67.44 | 38.46 | 67.88 | 59.00 |
> | IAgent     | Claude-3.7-Sonnet | 84.62        | 72.41        | 50.00        | 72.73     | 68.50       |
>
> We have updated Table 1. As the experimental results are numerous, we suggest you check our updated table for easy comparison.
>
> The experimental results show: 1. When using the same model (GPT-4o or GPT-4.1), IAgent shows an improvement of approximately 8-11 percentage points compared to smolagents. 2. When experimenting with Claude-3.7-Sonnet, IAgent still maintains a lead of at least 3 percentage points compared to other baselines. 3. Even when switching to a smaller model, IAgent can still achieve impressive performance, even exceeding some frameworks based on fine-tuned models.
>
> Finally, we realized that the discussion regarding combined models is an engineering-level issue. It is impossible to objectively compare the differences between frameworks. Therefore, we have placed the content regarding this aspect in Appendix C.

---

> > ### Author Response · Authors · 2025-11-23
> > **Part 2**
> >
> > **W2:** While there is an overall improvement over Smalagents, there is a significant degradation in performance on level-3, representing the more difficult subset. This brings into question whether the paper's method is mainly effective for the more easier deep research questions. I would have expected the opposite, since the proposed CoderAgent and reasoning-verification should be more effective for the harder tasks. The authors should consider benchmarking on BrowseComp, which has been considered much more harder benchmark for deep research.
> >
> > **A2:** Thank you for pointing this out. This is indeed a strange phenomenon, so we further examined the trajectories and code of IAgent.  We found that the original smolagents repository does not have a context compression mechanism for the SearchAgent.  They control its context length by limiting the number of action steps of the SearchAgent.  When the maximum number of action steps is reached, the SearchAgent is forced to summarize the existing information and report back to the ManagerAgent.  When using GPT-4.1, based on its larger context window, we extended the SearchAgent's action steps from 20 to 30.  Level-3 tasks are more complex, requiring the SearchAgent to execute more action steps.  More action steps lead to a significantly increased rate of context accumulation.  We incorrectly estimated the impact of context accumulation.  When using GPT-4.1, the SearchAgent often exceeded the context length limit and was interrupted before reaching the maximum number of steps (30).  After restoring the max steps to the default of 20, GPT-4.1 was able to complete the tasks stably.  Based on this setting, we supplemented the evaluation of GPT-4.1 on the GAIA benchmark level-3, achieving an accuracy of 38.46%.  It is important to emphasize that all supplementary experiments in [W1] strictly followed this configuration (max steps=20) to ensure comparability and consistency of the results.
> >
> > In addition, we have supplemented experiments for IAgent on the BrowseComp benchmark. The results are as follows:
> >
> > | Method                    | Model                        | Browsecomp |
> > | ------------------------- | ---------------------------- | ---------- |
> > | **LLM-based ReAct Agent** |                              |            |
> > |                           | GPT-4.1                      | 7.9%       |
> > |                           | Claude-3.7                   | 4.8%       |
> > |                           | OpenAI-o1                    | 14.3%      |
> > | **Deep Research Agent**   |                              |            |
> > |                           | OAgents(Claude-3.7-Sonnet)   | 22.2%      |
> > |                           | BrowseMaster(DeepSeek-R1)    | 30.0%      |
> > |                           | WebExplorer(Claude-4-Sonnet) | 12.2%      |
> > |                           | IAgent(Claude-3.7-Sonnet)    | 26.0%      |
> >
> > IAgent achieves exceptional performance on this benchmark, surpassing the Claude-4-Sonnet-powered WebExplorer. While IAgent trails the reasoning-model-driven BrowseMaster, this outcome is attributed to the high density of complex reasoning tasks within BrowseComp, which imposes significant demands on the reasoning capabilities of the foundation models. This observation is further corroborated by the ReAct Agent experiments, where the reasoning-enhanced OpenAI-o1 significantly outperforms general-purpose models such as GPT-4.1 and Claude-3.7-Sonnet.

---

> ### Author Response · Authors · 2025-11-23
> **Part 3**
>
> **W3:** The approach in the paper looks highly hand-stitched and only relying on closed-source models, making it unclear how to pursue further improvements with training. The authors should consider showing performance with an open-source model like GPT-OSS-20B or Qwen3-32B. Would be great if they can further improve performance of these models with an approach like SFT over closed-source model trajectories.
>
> **A3:** Thank you for your suggestion. As mentioned in [W1], we have supplemented experiments for IAgent based on QwQ-32B. The results are as follows:
>
> | Method | Model   | GAIA level 1 | GAIA level 2 | GAIA level 3 | GAIA avg. | WebWalkerQA |
> | :----- | :------ | :----------- | :----------- | :----------- | :-------- | :---------- |
> | IAgent | QwQ-32B | 52.8         | 48.8         | 19.2         | 45.45     | 47.50       |
>
> IAgent's performance not only surpasses frameworks also using QwQ-32B but even exceeds some frameworks based on fine-tuned models. This demonstrates that our framework generalizes well to mid-sized models.
>
> We also agree that fine-tuning based on closed-source model trajectories could further enhance the performance of these open-source models. However, due to resource constraints, we do not currently plan to pursue this. Our objective is to demonstrate that integrating our modules into any existing deep research framework can lead to reduced token consumption, lower latency, and higher accuracy. Furthermore, we posit that the inertia bias of LLMs is not limited to the domain of deep research. We hope our approach offers insights for addressing similar bias issues in other fields. For instance, in code generation, where agents may fall into infinite loops due to an initial incorrect debugging strategy, we raise the question: can isolating specific parts of the context help correct the agent's erroneous reasoning path?
>
> **W4:** The related work section needs more revision, specifically to incorporate discussion on how the proposed method differs from existing multi-agent methods. The related work currently just reads like a summary of prior work in each sub-category.
>
> **A4:** Thanks for the suggestion. We have reorganized the related work section to explicitly highlight the differences between our approach and current deep research works. Specifically: 1. While most frameworks aim to correct action trajectories after errors occur, IAgent is the first to identify the specific issue of search noise in deep research frameworks and address it directly. 2. Although many frameworks strive to mitigate hallucinations and noise in agent systems, they typically rely on external knowledge bases, such as KnowAgent and Agent KB. In contrast, IAgent avoids dependency on external knowledge bases. Instead, it leverages the self-correction capabilities of LLMs to identify issues through mechanisms like the isolation-based stepwise validation module. 3. Even though previous agent frameworks have adopted code as an action representation, they have not specifically tailored the web browsing components for the deep research domain. Our work demonstrates the effectiveness of code-driven deep browsing for solving web search tasks that require multi-step iteration.
>
> **Questions**
>
> **Q1:** While token-count cost is compared, how does the comparison of inference time look like for IAgent vs Smolagents when using the same LLM?
>
> **A1:** Thank you for the question. Following your suggestion, we calculated the average runtime per question for IAgent vs. smolagents when using GPT-4o. The result are as follows:
>
> | Method        | Model  | GAIA     | WebWalker |
> | :------------ | :----- | :------- | :-------- |
> | IAgent        | GPT-4o | 8.17 min | 2.59 min  |
> | smolagents DR | GPT-4o | 9.70 min | 3.22 min  |
>
> IAgent's average runtime on both benchmarks is lower than that of smolagents DR. Currently, the second stage (Step-by-Step Reasoning Validation) of our isolation-based stepwise validation module runs serially; if run in parallel, the runtime would be even shorter. We have now added the runtime analysis to the revised paper.
>
> In addition, in our initial version, we only selected only 20 queries to compare costs. We have now increased the number of queries to 50. The updated token-count cost is as follows:
>
> | Method        | Model  | Input Tokens             | Output Tokens        | Total Tokens             |
> | :------------ | :----- | :----------------------- | :------------------- | :----------------------- |
> | smolagents DR | GPT-4o | 217.6k                   | 2.2k                 | 219.8k                   |
> | IAgent        | GPT-4o | 144.1k $\downarrow$33.8% | 3.2k $\uparrow$45.5% | 147.3k $\downarrow$33.0% |
>
> The conclusion remains consistent with our previous findings: IAgent significantly reduces token consumption.
>
> ---
>
> We hope our response addresses your concern.

---

### Note · Authors · 2025-12-02

I have read and agree with the venue's withdrawal policy on behalf of myself and my co-authors.